# RAD: Training an End-to-End Driving Policy via Large-Scale 3DGS-based Reinforcement Learning

**Hao Gao**[1,◇] **Shaoyu Chen**[1,2,†] **Bo Jiang**[1] **Bencheng Liao**[1] **Yiang Shi**[1]
**Xiaoyang Guo**[2] **Yuechuan Pu**[2] **Haoran Yin**[2] **Xiangyu Li**[2] **Xinbang Zhang**[2]
**Ying Zhang**[2] **Wenyu Liu**[1] **Qian Zhang**[2] **Xinggang Wang**[1,✉]
[1] Huazhong University of Science & Technology    [2] Horizon Robotics

## Abstract

Existing end-to-end autonomous driving (AD) algorithms typically follow the Imitation Learning (IL) paradigm, which faces challenges such as causal confusion and an open-loop gap. In this work, we propose **RAD**, a 3DGS-based closed-loop **R**einforcement Learning (RL) framework for end-to-end **A**utonomous **D**riving. By leveraging 3DGS techniques, we construct a photorealistic digital replica of the real physical world, enabling the AD policy to extensively explore the state space and learn to handle out-of-distribution scenarios through large-scale trial and error. To enhance safety, we design specialized rewards to guide the policy in effectively responding to safety-critical events and understanding real-world causal relationships. To better align with human driving behavior, we incorporate IL into RL training as a regularization term. We introduce a closed-loop evaluation benchmark consisting of diverse, previously unseen 3DGS environments. Compared to IL-based methods, RAD achieves stronger performance in most closed-loop metrics, particularly exhibiting a $3\times$ lower collision rate. Abundant closed-loop results are presented in the supplementary material. Code is available at `https://github.com/hustvl/RAD` for facilitating future research.

## 1  Introduction

End-to-end autonomous driving (AD) is currently a trending topic in both academia and industry. It replaces a modularized pipeline with a holistic one by directly mapping sensory inputs to driving actions, offering advantages in system simplicity and generalization ability. Most existing end-to-end AD algorithms [1, 2, 3, 4, 5, 6, 7, 8] follow the Imitation Learning (IL) paradigm, which trains a neural network to mimic human driving behavior. However, despite their simplicity, IL-based methods face significant challenges in real-world deployment.

One key issue is **causal confusion**. IL trains a network to replicate human driving policies by learning from demonstrations. However, this paradigm primarily captures correlations rather than causal relationships between observations (states) and actions. As a result, IL-trained policies may struggle to identify the true causal factors behind planning decisions, leading to shortcut learning [9], *e.g.*, merely extrapolating future trajectories from historical ones [10, 11]. Furthermore, since IL training data predominantly consists of common driving behaviors and does not adequately cover long-tailed distributions, IL-trained policies tend to converge to trivial solutions, lacking sufficient sensitivity to safety-critical events such as collisions.

Another major challenge is the **gap between open-loop training and closed-loop deployment**. IL policies are trained in an open-loop manner using well-distributed driving demonstrations. However,

---

◇ The work was done when Hao Gao (`g_hao@hust.edu.cn`) was an intern of Horizon Robotics.

† Project lead (`shaoyu.chen@horizon.auto`). ✉ Corresponding author (`xgwang@hust.edu.cn`).

39th Conference on Neural Information Processing Systems (NeurIPS 2025).

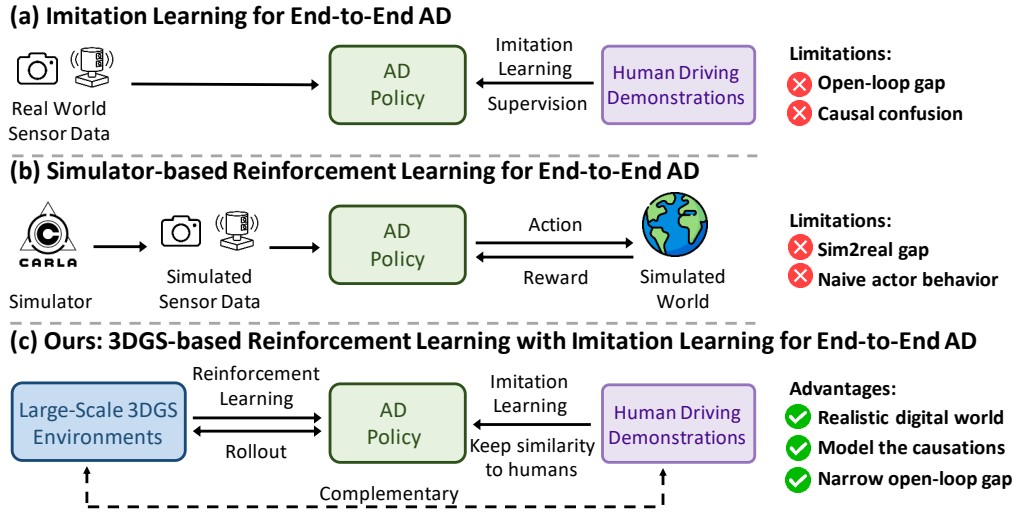

Figure 1: Different training paradigms of end-to-end autonomous driving (AD).

real-world driving is a closed-loop process where minor trajectory errors at each step accumulate over time, leading to compounding errors and out-of-distribution scenarios. IL-trained policies often struggle in these unseen situations, raising concerns about their robustness.

A straightforward solution to these problems is to perform closed-loop Reinforcement Learning (RL) training, which requires a driving environment that can interact with the AD policy. However, using real-world driving environments for closed-loop training poses prohibitive safety risks and operational costs. Simulated driving environments with sensor data simulation capabilities [12, 13] (which are required for end-to-end AD) are typically built on game engines [14, 15] but fail to provide realistic sensor simulation results.

In this work, we establish a 3DGS-based [16] closed-loop RL training paradigm. Leveraging 3DGS techniques, we construct a photorealistic digital replica of the real world, where the AD policy can extensively explore the state space and learn to handle out-of-distribution situations through large-scale trial and error. To ensure effective responses to safety-critical events and a better understanding of real-world causations, we design specialized safety-related rewards. However, RL training presents several critical challenges, which this paper addresses.

One significant challenge is the **Human Alignment Problem**. The exploration process in RL can lead to policies that deviate from human-like behavior, disrupting the smoothness of the action sequence. To address this, we incorporate imitation learning as a regularization term during RL training, helping to maintain similarity to human driving behavior. As illustrated in Fig. 1, RL and IL work together to optimize the AD policy: RL enhances IL by addressing causation and the open-loop gap, while IL improves RL by ensuring better human alignment.

Another major challenge is the **Sparse Reward Problem**. RL often suffers from sparse rewards and slow convergence. To alleviate this issue, we introduce dense auxiliary objectives related to collisions and deviations, which help constrain the full action distribution. Additionally, we streamline and decouple the action space to reduce the exploration cost associated with RL.

To validate the effectiveness of our approach, we construct a closed-loop evaluation benchmark comprising diverse, unseen 3DGS environments. Our method, RAD, outperforms IL-based approaches across most closed-loop metrics, notably achieving a collision rate that is $3\times$ lower.

The contributions of this work are summarized as follows:

- We propose the first 3DGS-based RL framework for training end-to-end AD policy. The reward, action space, optimization objective, and interaction mechanism are specially designed to enhance training efficiency and effectiveness.

- We propose to combine RL and IL to synergistically optimize the end-to-end AD policy. RL complements IL by modeling the causations and narrowing the open-loop gap, while IL complements RL in terms of human alignment.

- We validate the effectiveness of RAD on a closed-loop evaluation benchmark consisting of diverse, unseen 3DGS environments. RAD achieves stronger performance in closed-loop evaluation, particularly a $3\times$ lower collision rate, compared to IL-based methods.

## 2 Related Work

**Dynamic Scene Reconstruction.** Implicit neural representations have been widely used in novel view synthesis and dynamic scene reconstruction, as in UniSim [17], MARS [18], and NeuRAD [19], which leverage neural scene graphs for structured decomposition. However, their slow rendering speeds hinder real-time applications. Several recent works [20, 21, 22] have demonstrated the effectiveness of 3D Gaussian Splatting (3DGS) [16] for dynamic urban scene reconstruction. While prior works [17, 18, 22] primarily utilize reconstructed scenes for closed-loop evaluation, we go a step further by incorporating scenes reconstructed via 3DGS into the RL training loop to enhance policy learning.

**End-to-End Imitation Learning for Autonomous Driving.** Recent advances in learning-based planning have shown strong potential, driven by large-scale data. UniAD [1] integrates multiple perception tasks to boost planning, while VAD [2] improves efficiency with compact vectorized representations. Follow-up works [23, 4, 3, 10, 24, 25, 26, 27, 7] enhance the single-trajectory paradigm, whereas VADv2 [6] and Hydra-MDP [5] shift towards multi-modal planning with improved scoring. DiffusionDrive [8] proposes a truncated diffusion policy that denoises an anchored Gaussian distribution to a multi-mode driving action distribution. While existing methods largely follow an IL paradigm, we enhance it by incorporating closed-loop RL to further optimize the policy.

**Reinforcement Learning.** Reinforcement Learning is a promising technique that has not been fully explored. AlphaGo [28] and AlphaGo Zero [29] have demonstrated the power of Reinforcement Learning in the game of Go. Recently, OpenAI O1 [30] and Deepseek-R1 [31] have leveraged Reinforcement Learning to develop reasoning abilities. RL has also been applied in autonomous driving [32, 33, 34, 35, 36], though often using non-photorealistic simulators (e.g., CARLA [12]) or requiring perfect perception inputs. To the best of our knowledge, RAD is the first to train an end-to-end autonomous driving agent using RL in a photorealistic 3DGS environment.

**Combining Imitation and Reinforcement Learning.** To combine the sample efficiency of IL with the robustness of RL, CADRE [37] and CIRL [38] adopt a two-stage pipeline, first imitating expert demonstrations and then fine-tuning policies with RL to improve success rates in basic and complex CARLA scenarios. Lu *et al.* [35] and Huang *et al.* [39] instead perform joint IL and RL optimization, where IL constrains RL exploration to maintain human-like behaviors while improving robustness and rare-case success. Although these approaches demonstrate the benefits of combining IL and RL, they are still limited to non-photorealistic simulators or rely on structured BEV representations and virtual sensor data. In contrast, RAD is the first to jointly integrate IL and RL within a 3DGS-based photorealistic digital-twin environment, enabling fully end-to-end policy learning from real-world sensor inputs.

## 3 RAD

### 3.1 End-to-End Driving Policy

The overall framework of RAD is depicted in Fig. 2. RAD takes multi-view image sequences as input, transforms the sensor data into scene token embeddings, outputs the probabilistic distribution of actions, and samples an action to control the vehicle.

**BEV Encoder.** We first employ a BEV encoder [40] to transform multi-view image features from the perspective view to the Bird's Eye View (BEV), obtaining a feature map in the BEV space. This feature map is then used to learn instance-level map features and agent features.

**Map Head.** Then we utilize a group of map tokens [41, 42, 43] to learn the vectorized map elements of the driving scene from the BEV feature map, including lane centerlines, lane dividers, road boundaries, arrows, traffic signals, *etc*.

**Agent Head.** Besides, a group of agent tokens [44] is adopted to predict the motion information of other traffic participants, including location, orientation, size, speed, and multi-mode future trajectories.

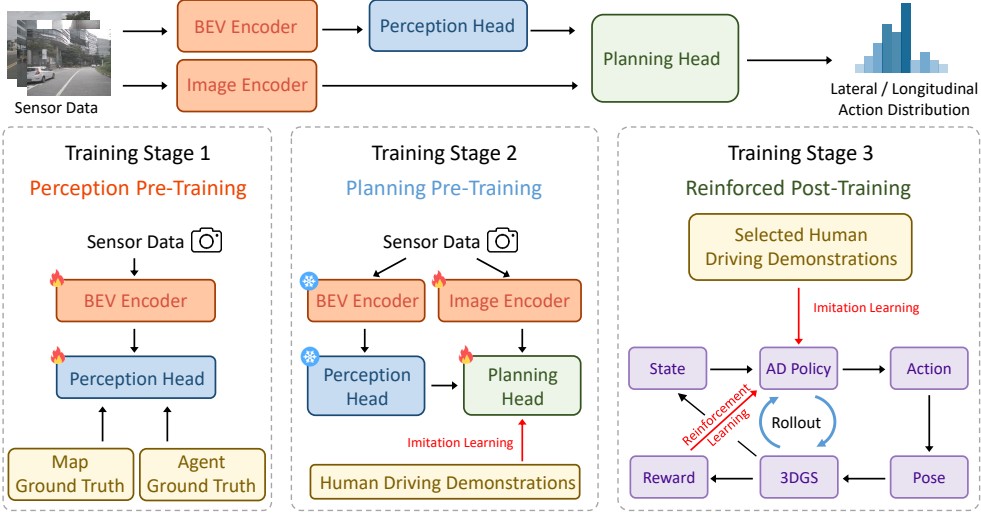

Figure 2: Overall framework of RAD. RAD takes a three-stage training paradigm. In the perception pre-training, ground-truths of map and agent are used to guide instance-level tokens to encode corresponding information. In the planning pre-training stage, large-scale driving demonstrations are used to initialize the action distribution. In the reinforced post-training stage, RL and IL synergistically fine-tune the AD policy.

**Image Encoder.** Apart from the above instance-level map and agent tokens, we use an individual image encoder [45, 46] to transform the original images into image tokens. These image tokens provide dense and rich scene information for planning, complementary to the instance-level tokens.

**Action Space.** We propose a decoupled discrete action representation by separating lateral and longitudinal actions. Each action is defined over a short 0.5-second horizon, assuming constant linear and angular velocities. This modeling assumption allows direct computation of control signals from the current action predictions. The resulting formulation reduces the dimensionality of the action space, facilitating more efficient policy optimization.

**Planning Head.** We use $E_{\text{scene}}$ to denote the scene representation, which consists of map tokens, agent tokens, and image tokens. We initialize a planning embedding denoted as $E_{\text{plan}}$. A cascaded Transformer decoder $\phi$ takes the planning embedding $E_{\text{plan}}$ as the query and the scene representation $E_{\text{scene}}$ as both key and value.

The output of the decoder $\phi$ is then combined with navigation information $E_{\text{navi}}$ and ego state $E_{\text{state}}$ to output the probabilistic distributions of the lateral action $a^x$ and the longitudinal action $a^y$:

$$
\begin{aligned}
\pi(a^x \mid s) &= \text{softmax}(\text{MLP}(\phi(E_{\text{plan}}, E_{\text{scene}}) \quad + E_{\text{navi}} + E_{\text{state}})), \\
\pi(a^y \mid s) &= \text{softmax}(\text{MLP}(\phi(E_{\text{plan}}, E_{\text{scene}}) \quad + E_{\text{navi}} + E_{\text{state}})),
\end{aligned}
\tag{1}
$$

where $E_{\text{plan}}$, $E_{\text{navi}}$, $E_{\text{state}}$, and the output of MLP are all of the same dimension ($1 \times D$).

The planning head also outputs the value functions $V_x(s)$ and $V_y(s)$, which estimate the expected cumulative rewards for the lateral and longitudinal actions, respectively:

$$
\begin{aligned}
V_x(s) &= \text{MLP}(\phi(E_{\text{plan}}, E_{\text{scene}}) + E_{\text{navi}} + E_{\text{state}}), \\
V_y(s) &= \text{MLP}(\phi(E_{\text{plan}}, E_{\text{scene}}) + E_{\text{navi}} + E_{\text{state}}).
\end{aligned}
\tag{2}
$$

The value functions are used in RL training (Sec. 3.5).

## 3.2 Training Paradigm

We adopt a three-stage training paradigm: perception pre-training, planning pre-training, and reinforced post-training, as shown in Fig. 2.

**Perception Pre-Training.** Information in the image is sparse and low-level. In the first stage, the map head and the agent head explicitly output map elements and agent motion information, which are supervised with ground-truth labels. Consequently, map tokens and agent tokens implicitly encode

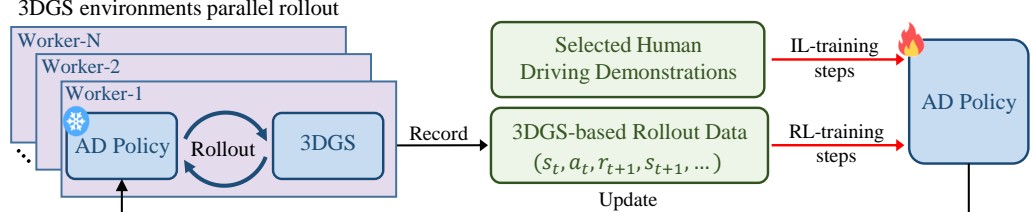

Figure 3: Post-training. $N$ workers parallelly run. The generated rollout data $(s_t, a_t, r_{t+1}, s_{t+1}, ...)$ are recorded in a rollout buffer. Rollout data and human driving demonstrations are used in RL- and IL-training steps to fine-tune the AD policy synergistically.

the corresponding high-level information. In this stage, we only update the parameters of the BEV encoder, the map head, and the agent head.

**Planning Pre-Training.** In the second stage, to prevent the unstable cold start of RL training, IL is first performed to initialize the probabilistic distribution of actions based on large-scale real-world driving demonstrations from expert drivers. In this stage, we only update the parameters of the image encoder and the planning head, while the parameters of the BEV encoder, map head, and agent head are frozen. The optimization objectives of perception tasks and planning tasks may conflict with each other. However, with the training stage and parameters decoupled, such conflicts are mostly avoided.

**Reinforced Post-Training.** In the reinforced post-training, RL and IL synergistically fine-tune the distribution. RL aims to guide the policy to be sensitive to critical risky events and adaptive to out-of-distribution situations. IL serves as the regularization term to keep the policy's behavior similar to that of humans.

We select a large number of risky, dense-traffic clips from collected driving demonstrations, and for each clip, train an independent 3DGS model to reconstruct it as a digital driving environment. As illustrated in Fig. 3, we deploy $N$ parallel workers. Each worker randomly samples a 3DGS environment, performs a rollout where the AD policy controls the ego vehicle to interact with the environment, and stores the resulting trajectories $(s_t, a_t, r_{t+1}, s_{t+1}, ...)$ in a shared buffer.

For policy optimization, we alternate between RL and IL steps. In RL steps, the Proximal Policy Optimization (PPO) [47] is used to update the policy from the rollout buffer. In IL steps, expert demonstrations are used for supervised updates. After a fixed number of steps, the updated policy is synchronized across all workers to mitigate distributional shift. During training, only the image encoder and planning head are updated, while the BEV encoder, map head, and agent head are kept frozen. The detailed RL design is introduced below.

### 3.3 Interaction Mechanism between AD Policy and 3DGS Environment

In the 3DGS environment, the ego vehicle acts according to the AD policy. Other traffic participants can be controlled in many manners: 1) using a smart agent model; 2) using Intelligent Driver Model (IDM) for route tracking; 3) log-replay with real-world data. In order to fully recover the real-world traffic flow, we use log-replay to control other traffic participants in experiments. A conventional kinematic bicycle model is employed to iteratively update the ego vehicle's pose at every $\Delta t$ seconds as follows:

$$x_{t+1}^w = x_t^w + v_t \cos\left(\psi_t^w\right)\Delta t, \; y_{t+1}^w = y_t^w + v_t \sin\left(\psi_t^w\right)\Delta t, \; \psi_{t+1}^w = \psi_t^w + \frac{v_t}{L}\tan\left(\delta_t\right)\Delta t, \quad (3)$$

where $x_t^w$ and $y_t^w$ denote the position of the ego vehicle relative to the world coordinate; $\psi_t^w$ is the heading angle that defines the vehicle's orientation with respect to the world $x$-coordinate; $v_t$ is the linear velocity of the ego vehicle; $\delta_t$ is the steering angle of the front wheels; and $L$ is the wheelbase, i.e., the distance between the front and rear axles.

During the rollout process, the AD policy outputs actions $(a_t^x, a_t^y)$ for a 0.5-second time horizon at time step $t$. We derive the linear velocity $v_t$ and steering angle $\delta_t$ based on $(a_t^x, a_t^y)$. Based on the kinematic model in Eq. 3, the pose of the ego vehicle in the world coordinate system is updated from $p_t = (x_t^w, y_t^w, \psi_t^w)$ to $p_{t+1} = (x_{t+1}^w, y_{t+1}^w, \psi_{t+1}^w)$.

Based on the updated $p_{t+1}$, the 3DGS environment computes the new ego vehicle's state $s_{t+1}$. The updated pose $p_{t+1}$ and state $s_{t+1}$ serve as the input for the next iteration of the inference process.

Table 1: Comparison of training strategies in Stage 3.

| Stage 3 | CR↓ | DCR↓ | SCR↓ | DR↓ | PDR↓ | HDR↓ | ADD↓ | Long. Jerk↓ | Lat. Jerk↓ |
|---|---|---|---|---|---|---|---|---|---|
| IL | 0.229 | 0.211 | 0.018 | 0.066 | **0.039** | 0.027 | **0.238** | **3.928** | 0.103 |
| RL | 0.143 | 0.128 | 0.015 | 0.080 | 0.065 | **0.015** | 0.345 | 4.204 | 0.085 |
| RL+IL | **0.089** | **0.080** | **0.009** | **0.063** | 0.042 | 0.021 | 0.257 | 4.495 | **0.082** |

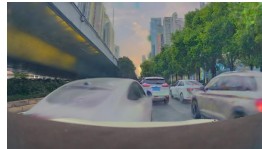 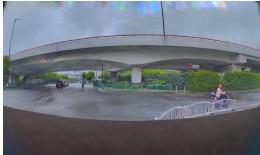 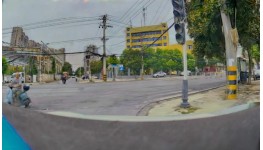 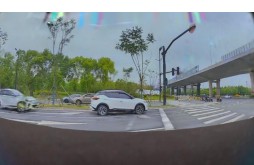

(1) Dynamic Collision      (2) Static Collision      (3) Positional Deviation      (4) Heading Deviation

Figure 4: Example diagram of four types of reward sources. (1) Collision with a dynamic obstacle ahead triggers $r_{\mathrm{dc}}$. (2) Hitting a static roadside obstacle incurs $r_{\mathrm{sc}}$. (3) Moving onto the curb triggers $r_{\mathrm{pd}}$. (4) Drifting toward the adjacent lane triggers $r_{\mathrm{hd}}$.

The 3DGS environment also generates rewards $\mathcal{R}$ (Sec. 3.4) according to multi-source information (including trajectories of other agents, map information, the expert trajectory of the ego vehicle, and the parameters of Gaussians), which are used to optimize the AD policy (Sec. 3.5).

## 3.4 Reward Modeling

The reward is the source of the training signal, which determines the optimization direction of RL. The reward function is designed to guide the ego vehicle's behavior by penalizing unsafe actions and encouraging alignment with the expert trajectory. It is composed of four reward components: (1) collision with dynamic obstacles, (2) collision with static obstacles, (3) positional deviation from the expert trajectory, and (4) heading deviation from the expert trajectory:

$$\mathcal{R} = \{r_{\mathrm{dc}}, r_{\mathrm{sc}}, r_{\mathrm{pd}}, r_{\mathrm{hd}}\}. \tag{4}$$

As illustrated in Fig. 4, these reward components are triggered under specific conditions. In the 3DGS environment, dynamic collision is detected if the ego vehicle's bounding box overlaps with the annotated bounding boxes of dynamic obstacles, triggering a negative reward $r_{\mathrm{dc}}$. Similarly, static collision is identified when the ego vehicle's bounding box overlaps with the Gaussians of static obstacles, resulting in a negative reward $r_{\mathrm{sc}}$. Positional deviation is measured as the Euclidean distance between the ego vehicle's current position and the closest point on the expert trajectory. A deviation beyond a predefined threshold $d_{\max}$ incurs a negative reward $r_{\mathrm{pd}}$. Heading deviation is calculated as the angular difference between the ego vehicle's current heading angle $\psi_t$ and the expert trajectory's matched heading angle $\psi_{\mathrm{expert}}$. A deviation beyond a threshold $\psi_{\max}$ results in a negative reward $r_{\mathrm{hd}}$.

Any of these events, including dynamic collision, static collision, excessive positional deviation, or excessive heading deviation, triggers immediate episode termination. Because after such events occur, the 3DGS environment typically generates noisy sensor data, which is detrimental to RL training.

## 3.5 Policy Optimization

In the closed-loop environment, the error in each single step accumulates over time. The aforementioned rewards are not only caused by the current action but also by the actions of the preceding steps. The rewards are propagated forward with Generalized Advantage Estimation (GAE) [48] to optimize the action distribution of the preceding steps.

Specifically, for each time step $t$, we store the current state $s_t$, action $a_t$, reward $r_t$, and the estimate of the value $V(s_t)$. Based on the decoupled action space, and considering that different rewards have different correlations to lateral and longitudinal actions, the reward $r_t$ is divided into lateral reward $r_t^x$ and longitudinal reward $r_t^y$:

$$r_t^x = r_t^{\mathrm{sc}} + r_t^{\mathrm{pd}} + r_t^{\mathrm{hd}}, \quad r_t^y = r_t^{\mathrm{dc}}. \tag{5}$$

Similarly, the value function $V(s_t)$ is decoupled into two components: $V_x(s_t)$ for the lateral dimension and $V_y(s_t)$ for the longitudinal dimension. These value functions estimate the expected cumulative rewards for the lateral and longitudinal actions, respectively. The advantage estimates $\hat{A}_t^x$ and $\hat{A}_t^y$ are computed using GAE (detailed in Appendix A.3), then decomposed into components aligned with reward types:

$$\hat{A}_t^x = \hat{A}_t^{\text{sc}} + \hat{A}_t^{\text{pd}} + \hat{A}_t^{\text{hd}}, \quad \hat{A}_t^y = \hat{A}_t^{\text{dc}}. \tag{6}$$

The policy $\pi_\theta$ is optimized via a modified PPO objective:

$$\mathcal{L}^{\text{PPO}}(\theta) = \mathcal{L}_x^{\text{PPO}}(\theta) + \mathcal{L}_y^{\text{PPO}}(\theta), \tag{7}$$

with independent clipping mechanisms for each dimension (full equations in Appendix A.3). The clipped objective prevents excessively large updates to the policy parameters $\theta$.

### 3.6 Auxiliary Objective

To address the sparse reward challenge in RL, we design directional auxiliary objectives that provide dense supervision for both longitudinal and lateral controls. These objectives adaptively adjust action probabilities based on real-time collision risk and trajectory deviation, offering more informative gradients during training.

Specifically, a dynamic collision auxiliary loss $\mathcal{L}_{\text{dc}}$ penalizes acceleration and encourages deceleration when past collisions with preceding vehicles are detected. A static collision auxiliary loss $\mathcal{L}_{\text{sc}}$ suppresses steering toward static obstacles and promotes avoidance behavior. For trajectory alignment, the positional deviation loss $\mathcal{L}_{\text{pd}}$ encourages lateral corrections toward the reference path, while the heading deviation loss $\mathcal{L}_{\text{hd}}$ promotes angular corrections to minimize orientation error. Additional implementation details are provided in Appendix A.4.

The composite optimization objective integrates the PPO policy gradient with auxiliary guidance:

$$\mathcal{L}(\theta) = \mathcal{L}^{\text{PPO}}(\theta) + \lambda_1 \mathcal{L}_{\text{dc}}(\theta) + \lambda_2 \mathcal{L}_{\text{sc}}(\theta) + \lambda_3 \mathcal{L}_{\text{pd}}(\theta) + \lambda_4 \mathcal{L}_{\text{hd}}(\theta), \tag{8}$$

where $\lambda_1$, $\lambda_2$, $\lambda_3$, and $\lambda_4$ are weighting coefficients that balance the contributions of each auxiliary objective.

Table 2: Impact of different reward components on performance.

| ID | Dyn. Col. | Sta. Col. | Pos. Dev. | Head. Dev. | CR↓ | DCR↓ | SCR↓ | DR↓ | PDR↓ | HDR↓ | ADD↓ | Long. Jerk↓ | Lat. Jerk↓ |
|---|---|---|---|---|---|---|---|---|---|---|---|---|---|
| 1 | ✓ | | | | 0.172 | 0.154 | 0.018 | 0.092 | 0.033 | 0.059 | 0.259 | 4.211 | 0.095 |
| 2 | | ✓ | ✓ | ✓ | 0.238 | 0.217 | 0.021 | 0.090 | 0.045 | 0.045 | **0.241** | 3.937 | 0.098 |
| 3 | ✓ | | ✓ | ✓ | 0.146 | 0.128 | 0.018 | 0.060 | **0.030** | 0.030 | 0.263 | 3.729 | 0.083 |
| 4 | ✓ | ✓ | | ✓ | 0.151 | 0.142 | 0.009 | 0.069 | 0.042 | 0.027 | 0.303 | 3.938 | 0.079 |
| 5 | ✓ | ✓ | ✓ | | 0.166 | 0.157 | 0.009 | **0.048** | 0.036 | **0.012** | 0.243 | **3.334** | **0.067** |
| 6 | ✓ | ✓ | ✓ | ✓ | **0.089** | **0.080** | **0.009** | 0.063 | 0.042 | 0.021 | 0.257 | 4.495 | 0.082 |

Table 3: Impact of different auxiliary objectives on performance.

| ID | PPO Obj. | Dyn. Col. Aux. Obj. | Sta. Col. Aux. Obj. | Pos. Dev. Aux. Obj. | Head. Dev. Aux. Obj. | CR↓ | DCR↓ | SCR↓ | DR↓ | PDR↓ | HDR↓ | ADD↓ | Long. Jerk↓ | Lat. Jerk↓ |
|---|---|---|---|---|---|---|---|---|---|---|---|---|---|---|
| 1 | ✓ | | | | | 0.249 | 0.223 | 0.026 | 0.077 | 0.047 | 0.030 | 0.266 | 4.209 | 0.104 |
| 2 | ✓ | ✓ | | | | 0.178 | 0.163 | 0.015 | 0.151 | 0.101 | 0.050 | 0.301 | 3.906 | 0.085 |
| 3 | ✓ | | ✓ | ✓ | ✓ | 0.137 | 0.125 | 0.012 | 0.157 | 0.145 | 0.012 | 0.296 | **3.419** | **0.071** |
| 4 | ✓ | ✓ | | ✓ | ✓ | 0.169 | 0.151 | 0.018 | 0.075 | 0.042 | 0.033 | 0.254 | 4.450 | 0.098 |
| 5 | ✓ | ✓ | ✓ | | ✓ | 0.149 | 0.134 | 0.015 | 0.063 | 0.057 | **0.006** | 0.324 | 3.980 | 0.086 |
| 6 | ✓ | ✓ | ✓ | ✓ | | 0.128 | 0.119 | 0.009 | 0.066 | 0.030 | 0.036 | **0.254** | 4.102 | 0.092 |
| 7 | | ✓ | ✓ | ✓ | ✓ | 0.187 | 0.175 | 0.012 | 0.077 | 0.056 | 0.021 | 0.309 | 5.014 | 0.112 |
| 8 | ✓ | ✓ | ✓ | ✓ | ✓ | **0.089** | **0.080** | **0.009** | **0.063** | **0.042** | 0.021 | 0.257 | 4.495 | 0.082 |

## 4 Experiments

### 4.1 Experimental Settings

**Dataset and Benchmark.** We collect 2000 hours of expert driving demonstrations in real-world conditions and generate map and agent annotations via an automated pipeline for perception pre-training. Ego-vehicle odometry is employed for planning pre-training. For reinforcement learning, we

Table 4: Closed-loop quantitative comparisons with other IL-based methods on the 3DGS evaluation benchmark.

| Method | CR↓ | DCR↓ | SCR↓ | DR↓ | PDR↓ | HDR↓ | ADD↓ | Long. Jerk↓ | Lat. Jerk↓ |
|---|---|---|---|---|---|---|---|---|---|
| TransFuser [27] | 0.320 | 0.273 | 0.047 | 0.235 | 0.188 | 0.047 | 0.263 | 4.538 | 0.142 |
| VAD [2] | 0.335 | 0.273 | 0.062 | 0.314 | 0.255 | 0.059 | 0.304 | 5.284 | 0.550 |
| GenAD [3] | 0.341 | 0.299 | 0.042 | 0.291 | 0.160 | 0.131 | 0.265 | 11.37 | 0.320 |
| VADv2 [6] | 0.270 | 0.240 | 0.030 | 0.243 | 0.139 | 0.104 | 0.273 | 7.782 | 0.171 |
| RAD | **0.089** | **0.080** | **0.009** | **0.063** | **0.042** | **0.021** | **0.257** | **4.495** | **0.082** |

select 4305 real-world driving scenes covering diverse road types, traffic densities, and agent behaviors to ensure environmental diversity. These scenes are first reconstructed into 3DGS environments, from each of which a fixed-length 8-second clip is extracted. Among these clips, 3968 are used for RL training, and the other 337 are used as closed-loop evaluation benchmarks. More training details are in the Appendix A.5.

**Metric.** We evaluate the performance of the AD policy using nine key metrics. Dynamic Collision Ratio (DCR) and Static Collision Ratio (SCR) quantify the frequency of collisions with dynamic and static obstacles, respectively, with their sum represented as the Collision Ratio (CR). Positional Deviation Ratio (PDR) measures the ego vehicle's deviation from the expert trajectory with respect to position, while Heading Deviation Ratio (HDR) evaluates the ego vehicle's consistency to the expert trajectory with respect to the forward direction. The overall deviation is quantified by the Deviation Ratio (DR), defined as the sum of PDR and HDR. Average Deviation Distance (ADD) quantifies the mean closest distance between the ego vehicle and the expert trajectory before any collisions or deviations occur. Additionally, Longitudinal Jerk (Long. Jerk) and Lateral Jerk (Lat. Jerk) assess driving smoothness by measuring acceleration changes in the longitudinal and lateral directions. CR, DCR, and SCR mainly reflect the policy's safety, and ADD reflects the trajectory consistency between the AD policy and human drivers. More details are provided in Appendix A.6.

## 4.2 Ablation Study

To evaluate the impact of different design choices in RAD, we conduct three ablation studies. These studies highlight the importance of combining RL and IL, the role of different reward sources, and the effect of auxiliary objectives.

**Training Strategy Comparison.** We compare pure IL, pure RL, and our mixed IL+RL strategy in reinforced post-training (Tab. 1). Pure IL achieves low deviation (ADD 0.238) but suffers from high collision risk (CR 0.229). Pure RL improves safety (CR 0.143) but deviates more from expert behavior (ADD 0.345). Our IL+RL strategy achieves the best balance (CR 0.089, ADD 0.257), demonstrating improved safety without sacrificing behavioral fidelity.

**Reward Source Analysis.** We analyze the influence of different reward components (Tab. 2). Policies trained with only partial reward terms (e.g., ID 1, 2, 3, 4, 5) exhibit higher collision rates (CR) compared to the full reward setup (ID 6), which achieves the lowest CR (0.089) while maintaining a stable ADD (0.257). This demonstrates that a well-balanced reward function, incorporating all reward terms, effectively enhances both safety and trajectory consistency. Among the partial reward configurations, ID 2, which omits the dynamic collision reward term, exhibits the highest CR (0.238), indicating that the absence of this term significantly impairs the model's ability to avoid dynamic obstacles, resulting in a higher collision rate.

**Auxiliary Objective Analysis.** We examine the impact of auxiliary objectives (Tab. 3). Compared to the full auxiliary setup (ID 8), omitting any auxiliary objective increases CR, with a significant rise observed when all auxiliary objectives are removed. This highlights their collective role in enhancing safety. Notably, ID 1, which retains all auxiliary objectives but excludes the PPO objective, results in a CR of 0.187. This value is higher than that of ID 8, indicating that while auxiliary objectives help reduce collisions, they are most effective when combined with the PPO objective.

## 4.3 Comparisons with Existing Methods

As presented in Tab. 4, we compare RAD with other end-to-end autonomous driving methods in the proposed 3DGS-based closed-loop evaluation. For fair comparisons, all the methods are trained with the same amount of human driving demonstrations. The 3DGS environments for the RL training in

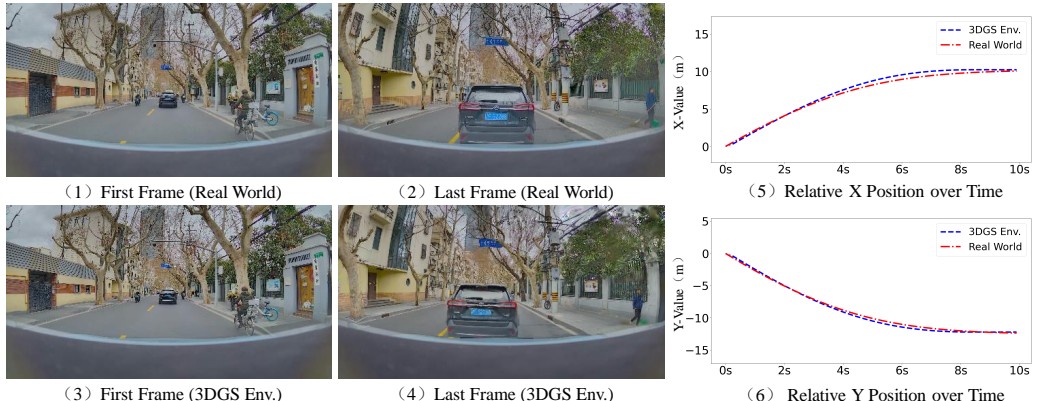

(1) First Frame (Real World)  (2) Last Frame (Real World)  (5) Relative X Position over Time

(3) First Frame (3DGS Env.)  (4) Last Frame (3DGS Env.)  (6) Relative Y Position over Time

Figure 5: Consistency analysis between 3DGS environment and real-world environment. We compare the driving behaviors of the same driving policy in both environments. Subfigure (1)–(4) present the first and last frames during closed-loop evaluation in both environments. Subfigure (5)-(6) depict the temporal evolution of the ego vehicle's position.

RAD are also based on these data. RAD achieves better performance compared to IL-based methods in most metrics. Especially in terms of CR, RAD achieves $3\times$ lower collision rate, demonstrating that RL helps the AD policy learn general collision avoidance ability.

## 4.4 Consistency Analysis

To demonstrate the consistency between 3DGS environment and real-world environment, we compare the driving behaviors of the same driving policy in both environments. As illustrated in Fig. 5, subfigure (1)–(4) present the first and last frames during closed-loop evaluation in both environments. Subfigure (5)-(6) depict the temporal evolution of the ego vehicle's position. Both qualitative and quantitative results indicate a high degree of behavioral consistency between the 3DGS environment and the real world environment. It shows 3DGS-based closed-loop evaluation can reflect the real-world driving performance.

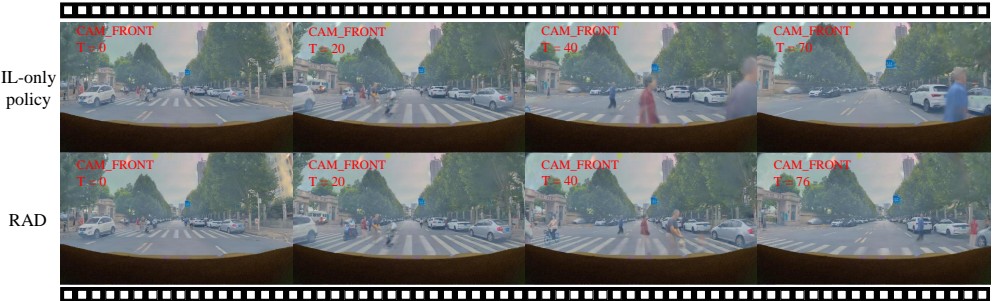

Figure 6: Closed-loop qualitative comparisons between the IL-only policy and RAD in a Yield to Pedestrians scenario. The IL-only policy fails to yield (Row 1), while RAD successfully yields to pedestrians (Row 2).

## 4.5 Qualitative Comparisons

We provide qualitative comparisons between the IL-only AD policy (without reinforced post-training) and RAD, as shown in Fig. 6. The IL-only method struggles in dynamic environments, frequently failing to avoid collisions with moving obstacles or manage complex traffic situations. In contrast, RAD consistently performs well, effectively avoiding dynamic obstacles and handling challenging tasks. These results highlight the benefits of closed-loop training in the hybrid method, which enables better handling of dynamic environments. Additional visualizations are included in Fig. 7.

# 5 Conclusion

In this work, we propose **RAD**, the first 3DGS-based closed-loop reinforcement learning framework for end-to-end autonomous driving. We combine RL and IL, with RL complementing IL to model the causations and narrow the open-loop gap, and IL complementing RL in terms of human alignment. Complemented by a targeted reward system and auxiliary objectives, RAD achieves 3× lower collision rates than state-of-the-art IL methods, with strong performance in challenging scenarios like unprotected left-turns and dense traffic.

**Limitations and future work.** The effect of 3DGS still has room for improvement, particularly in rendering non-rigid pedestrians, unobserved views, and low-light scenarios. Future works will focus on addressing these issues and scaling up RL to the next level.

**Acknowledgement**: This work was partially supported by National Natural Science Foundation of China (No. 62376102).

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

# A Technical Appendices

## A.1 3DGS Reconstruction and Rendering Optimization

To support closed-loop training, we extend the StreetGaussian [20] framework, focusing on enhancing rendering realism and geometric accuracy, particularly for high-fidelity rendering in off-trajectory views.

First, we employ mesh modeling to constrain the geometry of the road surface, constraining Gaussian spheres to the mesh surface to ensure precise road geometry from any viewpoint. Additionally, we model the sky separately to prevent confusion with foreground objects, improving rendering realism under complex lighting conditions.

For foreground objects (e.g., vehicles, pedestrians), we optimize their poses during training and incorporate depth and normal consistency as supervision signals to further enhance geometric reconstruction accuracy and surface detail fidelity. These optimizations significantly improve rendering quality in novel viewpoints, especially in dynamic scenes, where object motion trajectories and surface details remain consistent with real-world observations.

These improvements allow the 3DGS environment to more effectively support closed-loop training, providing a foundation of high realism and geometric accuracy for large-scale trial-and-error learning in autonomous driving strategies.

## A.2 Action Space Details

Here, we provide a more comprehensive explanation of the action space design. To ensure stable control and efficient learning, we define the action space over a short time horizon of 0.5 seconds. The ego vehicle's movement is modeled using discrete displacements in both the lateral and longitudinal directions.

**Lateral Displacement.** The lateral displacement, denoted as $a^x$, represents the vehicle's movement in the lateral direction over the 0.5-second horizon. We discretize this dimension into $N_x$ options, symmetrically distributed around zero to allow leftward and rightward movements, with an additional option to maintain the current trajectory. The set of possible lateral displacements is:

$$a^x \in \{d^x_{\min}, \ldots, 0, \ldots, d^x_{\max}\}. \tag{9}$$

In our implementation, we use $N_x = 61$, with $d^x_{\min} = -0.75$ m, $d^x_{\max} = 0.75$ m, and intermediate values sampled uniformly.

**Longitudinal Displacement.** The longitudinal displacement, denoted as $a^y$, represents the vehicle's movement in the forward direction over the 0.5-second horizon. Similar to the lateral component, we discretize this dimension into $N_y$ options, covering a range of forward displacements, including an option to maintain the current position:

$$a^y \in \{0, \ldots, d^y_{\max}\}. \tag{10}$$

In our setup, we use $N_y = 61$, with $d^y_{\max} = 15$m, and intermediate values sampled uniformly.

## A.3 Policy Optimization Details

**GAE Computation Details.** The advantage estimates $\hat{A}^x_t$ and $\hat{A}^y_t$ are then computed as follows:

$$
\begin{aligned}
\delta^x_t &= r^x_t + \gamma V_x(s_{t+1}) - V_x(s_t), \\
\delta^y_t &= r^y_t + \gamma V_y(s_{t+1}) - V_y(s_t), \\
\hat{A}^x_t &= \sum_{l=0}^{\infty} (\gamma\lambda)^l \delta^x_{t+l}, \\
\hat{A}^y_t &= \sum_{l=0}^{\infty} (\gamma\lambda)^l \delta^y_{t+l},
\end{aligned}
\tag{11}
$$

where $\delta_t^x$ and $\delta_t^y$ are the temporal difference errors for the lateral and longitudinal dimensions, $\gamma$ is the discount factor, and $\lambda$ is the GAE parameter that controls the trade-off between bias and variance.

**Complete PPO Objectives.** The full clipping objectives are:

$$
\begin{aligned}
\mathcal{L}_x^{\text{PPO}}(\theta) &= \mathbb{E}_t \left[ \min \left( \rho_t^x \hat{A}_t^x, \ \text{clip}(\rho_t^x, 1 - \epsilon_x, 1 + \epsilon_x) \hat{A}_t^x \right) \right], \\
\mathcal{L}_y^{\text{PPO}}(\theta) &= \mathbb{E}_t \left[ \min \left( \rho_t^y \hat{A}_t^y, \ \text{clip}(\rho_t^y, 1 - \epsilon_y, 1 + \epsilon_y) \hat{A}_t^y \right) \right],
\end{aligned}
\tag{12}
$$

where $\rho_t^x = \frac{\pi_\theta(a_t^x | s_t)}{\pi_{\theta_{\text{old}}}(a_t^x | s_t)}$ is the importance sampling ratio for the lateral dimension, $\rho_t^y = \frac{\pi_\theta(a_t^y | s_t)}{\pi_{\theta_{\text{old}}}(a_t^y | s_t)}$ is the importance sampling ratio for the longitudinal dimension, $\epsilon_x$ and $\epsilon_y$ are small constants that control the clipping range for the lateral and longitudinal dimensions, ensuring stable policy updates.

## A.4 Auxiliary Objectives Details

**Action Probability Decomposition.** The auxiliary objectives are designed to penalize undesirable behaviors by incorporating specific reward sources, including dynamic collisions, static collisions, positional deviations, and heading deviations. These objectives are computed based on the actions $a_t^{x,\text{old}}$ and $a_t^{y,\text{old}}$ selected by the old AD policy $\pi_{\theta_{\text{old}}}$ at time step $t$. To facilitate the evaluation of these actions, we separate the probability distribution of the action into four parts:

$$
\begin{aligned}
\Delta\pi_y^{\text{dec}} &= \sum_{a_t^y < a_t^{y,\text{old}}} \pi_\theta(a_t^y \mid s_t), \\
\Delta\pi_y^{\text{acc}} &= \sum_{a_t^y > a_t^{y,\text{old}}} \pi_\theta(a_t^y \mid s_t), \\
\Delta\pi_x^{\text{left}} &= \sum_{a_t^x < a_t^{x,\text{old}}} \pi_\theta(a_t^x \mid s_t), \\
\Delta\pi_x^{\text{right}} &= \sum_{a_t^x > a_t^{x,\text{old}}} \pi_\theta(a_t^x \mid s_t).
\end{aligned}
\tag{13}
$$

Here, $\Delta\pi_y^{\text{dec}}$ represents the total probability of deceleration actions, $\Delta\pi_y^{\text{acc}}$ represents the total probability of acceleration actions, $\Delta\pi_x^{\text{left}}$ represents the total probability of leftward steering actions, and $\Delta\pi_x^{\text{right}}$ represents the total probability of rightward steering actions.

**Dynamic Collision Auxiliary Objective.** The dynamic collision auxiliary objective adjusts the longitudinal control action $a_t^y$ based on the location of potential collisions relative to the ego vehicle. If a collision is detected ahead, the policy prioritizes deceleration actions ($a_t^y < a_t^{y,\text{old}}$); if a collision is detected behind, it encourages acceleration actions ($a_t^y > a_t^{y,\text{old}}$). To formalize this behavior, we define a directional factor $f_{\text{dc}}$:

$$
f_{\text{dc}} = \begin{cases} 1 & \text{if the collision is ahead,} \\ -1 & \text{if the collision is behind.} \end{cases}
\tag{14}
$$

The auxiliary objective for dynamic collision avoidance is defined as:

$$
\mathcal{L}_{\text{dc}}(\theta) = \mathbb{E}_t \left[ \hat{A}_t^{\text{dc}} \cdot f_{\text{dc}} \cdot (\Delta\pi_y^{\text{dec}} - \Delta\pi_y^{\text{acc}}) \right],
\tag{15}
$$

where $\hat{A}_t^{\text{dc}}$ is the advantage estimate for dynamic collision avoidance.

**Static Collision Auxiliary Objective.** The static collision auxiliary objective adjusts the steering control action $a_t^x$ based on the proximity to static obstacles. If the static obstacle is detected on the left side, the policy promotes rightward steering actions ($a_t^x > a_t^{x,\text{old}}$); if the static obstacle is detected on the right side, it promotes leftward steering actions ($a_t^x < a_t^{x,\text{old}}$). To formalize this behavior, we define a directional factor $f_{\text{sc}}$:

$$
f_{\text{sc}} = \begin{cases} 1 & \text{if the static obstacle is on the left,} \\ -1 & \text{if the static obstacle is on the right.} \end{cases}
\tag{16}
$$

The auxiliary objective for static collision avoidance is defined as:

$$\mathcal{L}_{\text{sc}}(\theta) = \mathbb{E}_t \left[ \hat{A}_t^{\text{sc}} \cdot f_{\text{sc}} \cdot (\Delta\pi_x^{\text{right}} - \Delta\pi_x^{\text{left}}) \right], \tag{17}$$

where $\hat{A}_t^{\text{sc}}$ is the advantage estimate for static collision avoidance.

**Positional Deviation Auxiliary Objective.** The positional deviation auxiliary objective adjusts the steering control action $a_t^x$ based on the ego vehicle's lateral deviation from the expert trajectory. If the ego vehicle deviates leftward, the policy promotes rightward corrections ($a_t^x > a_t^{x,\text{old}}$); if it deviates rightward, it promotes leftward corrections ($a_t^x < a_t^{x,\text{old}}$). We formalize this with a directional factor $f_{\text{pd}}$:

$$f_{\text{pd}} = \begin{cases} 1 & \text{if ego vehicle deviates leftward,} \\ -1 & \text{if ego vehicle deviates rightward.} \end{cases} \tag{18}$$

The auxiliary objective for positional deviation correction is:

$$\mathcal{L}_{\text{pd}}(\theta) = \mathbb{E}_t \left[ \hat{A}_t^{\text{pd}} \cdot f_{\text{pd}} \cdot (\Delta\pi_x^{\text{right}} - \Delta\pi_x^{\text{left}}) \right], \tag{19}$$

where $\hat{A}_t^{\text{pd}}$ estimates the advantage of trajectory alignment.

**Heading Deviation Auxiliary Objective.** The heading deviation auxiliary objective adjusts the steering control action $a_t^x$ based on the angular difference between the ego vehicle's current heading and the expert's reference heading. If the ego vehicle deviates counterclockwise, the policy promotes clockwise corrections ($a_t^x > a_t^{x,\text{old}}$); if it deviates clockwise, it promotes counterclockwise corrections ($a_t^x < a_t^{x,\text{old}}$). To formalize this behavior, we define a directional factor $f_{\text{hd}}$:

$$f_{\text{hd}} = \begin{cases} 1 & \text{if ego vehicle deviates clockwise,} \\ -1 & \text{if ego vehicle deviates counterclockwise.} \end{cases} \tag{20}$$

The auxiliary objective for heading deviation correction is then defined as:

$$\mathcal{L}_{\text{hd}}(\theta) = \mathbb{E}_t \left[ \hat{A}_t^{\text{hd}} \cdot f_{\text{hd}} \cdot (\Delta\pi_x^{\text{right}} - \Delta\pi_x^{\text{left}}) \right], \tag{21}$$

where $\hat{A}_t^{\text{hd}}$ is the advantage estimate for heading alignment.

## A.5 Implementation Details

In this section, we summarize the training settings, configurations, and hyperparameters used in our approach.

**Planning Pre-Training.** The action space is discretized using predefined anchors $\mathcal{A} = \{(a_i^x, a_j^y)\}_{i=1,j=1}^{N_x,N_y}$. Each anchor corresponds to a specific steering-speed combination within the 0.5-second planning horizon. Given the ground truth vehicle position at $t = 0.5$ s denoted as $p_{\text{gt}} = (p_{\text{gt}}^x, p_{\text{gt}}^y)$, we implement normalized nearest-neighbor matching over predefined anchor positions:

$$\begin{aligned} \hat{i} &= \arg\min_i \left\| \frac{\mathbf{a}_i^x - d_{\min}^x}{d_{\max}^x - d_{\min}^x} - \frac{p_{\text{gt}}^x - d_{\min}^x}{d_{\max}^x - d_{\min}^x} \right\|_2, \\ \hat{j} &= \arg\min_j \left\| \frac{\mathbf{a}_j^y - 0}{d_{\max}^y - 0} - \frac{p_{\text{gt}}^y - 0}{d_{\max}^y - 0} \right\|_2. \end{aligned} \tag{22}$$

Based on the matched anchor indices $(\hat{i}, \hat{j})$, we formulate the imitation learning objective as a dual focal loss [49]:

$$\mathcal{L}_{\text{IL}} = \mathcal{L}_{\text{focal}}(\pi(a^x \mid s), \hat{i}_t) + \mathcal{L}_{\text{focal}}(\pi(a^y \mid s), \hat{j}_t), \tag{23}$$

where $\mathcal{L}_{\text{focal}}$ is focal loss for discrete action classification.

**Reinforced Post-Training.** During the training process, we use a cycle where reinforcement learning (RL) and imitation learning (IL) alternate. In each full cycle, we run four rounds of RL training, followed by one round of IL training.

Each RL training round consists of 320 iterations. Our clips are 8 seconds long with a frame rate of 10Hz, meaning that 320 iterations cover four clips of data. We employ a sliding window mechanism that holds four clips of data, updated in a first-in, first-out manner. In each RL training round, data from one new clip is collected and the sliding window is updated accordingly. We find that an RL-to-IL ratio of $(320 \times 4) : 320 = 4 : 1$ yields the best results.

**Training configurations.** We provide detailed hyperparameters for the two main stages, Planning Pre-Training and Reinforced Post-Training, in Tab. 5 and Tab. 6, respectively.

Table 5: Hyperparameters used in RAD Planning Pre-Training stage.

| config | Planning Pre-Training |
|---|---|
| learning rate | 1e-4 |
| learning rate schedule | cosine decay |
| optimizer | AdamW [50, 51] |
| optimizer hyper-parameters | $\beta_1, \beta_2, \epsilon = 0.9, 0.999, 1e\text{-}8$ |
| weight decay | 1e-4 |
| batch size | 512 |
| training steps | 30k |
| planning head dim | 256 |
| Traning GPU | 128 RTX4090 |

Table 6: Hyperparameters used in RAD Reinforced Post-Training stage.

| config | Reinforced Post-Training |
|---|---|
| learning rate | 5e-6 |
| learning rate schedule | cosine decay |
| optimizer | AdamW [50, 51] |
| optimizer hyper-parameters | $\beta_1, \beta_2, \epsilon = 0.9, 0.999, 1e\text{-}8$ |
| weight decay | 1e-4 |
| RL worker number | 32 |
| RL batch size | 32 |
| IL batch size | 128 |
| GAE parameter | $\gamma = 0.9, \lambda = 0.95$ |
| clipping thresholds | $\epsilon_x = 0.1, \epsilon_y = 0.2$ |
| deviation threshold | $d_{\max} = 2.0m, \psi_{\max} = 40°$ |
| planning head dim | 256 |
| value function dim | 256 |
| Traning GPU | 32 RTX4090 |

## A.6 Metric Details

We evaluate the performance of the autonomous driving policy using nine key metrics.

**Dynamic Collision Ratio (DCR).** DCR quantifies the frequency of collisions with dynamic obstacles. It is defined as:

$$DCR = \frac{N_{dc}}{N_{total}}, \tag{24}$$

where $N_{dc}$ is the number of clips in which collisions with dynamic obstacles occur, and $N_{total}$ is the total number of clips.

**Static Collision Ratio (SCR).** SCR measures the frequency of collisions with static obstacles and is defined as:

$$SCR = \frac{N_{sc}}{N_{total}}, \tag{25}$$

where $N_{sc}$ is the number of clips with static obstacle collisions.

**Collision Ratio (CR).** CR represents the total collision frequency, given by:

$$CR = DCR + SCR. \tag{26}$$

**Positional Deviation Ratio (PDR).** PDR evaluates the ego vehicle's adherence to the expert trajectory in terms of position. It is defined as:

$$PDR = \frac{N_{pd}}{N_{total}}, \tag{27}$$

where $N_{pd}$ is the number of clips in which the positional deviation exceeds a predefined threshold.

**Heading Deviation Ratio (HDR).** HDR assesses orientation accuracy by computing the proportion of clips where heading deviations surpass a predefined threshold:

$$HDR = \frac{N_{hd}}{N_{total}}, \tag{28}$$

where $N_{hd}$ is the number of clips where the heading deviation exceeds the threshold.

**Deviation Ratio (DR).** captures the overall deviation from the expert trajectory, given by:

$$DR = PDR + HDR. \tag{29}$$

**Average Deviation Distance (ADD).** ADD quantifies the mean closest distance between the ego vehicle and the expert trajectory during time steps when no collisions or deviations occur. It is defined as:

$$ADD = \frac{1}{T_{safe}} \sum_{t=1}^{T_{safe}} d_{min}(t), \tag{30}$$

where $T_{safe}$ represents the total number of time steps in which the ego vehicle operates without collisions or deviations, and $d_{min}(t)$ denotes the minimum distance between the ego vehicle and the expert trajectory at time step $t$.

Finally, **Longitudinal Jerk (Long. Jerk)** and **Lateral Jerk (Lat. Jerk)** quantify the smoothness of vehicle motion by measuring acceleration changes. Longitudinal jerk is defined as:

$$J_{long} = \frac{d^2 v_{long}}{dt^2}, \tag{31}$$

where $v_{long}$ represents the longitudinal velocity. Similarly, lateral jerk is defined as:

$$J_{lat} = \frac{d^2 v_{lat}}{dt^2}, \tag{32}$$

where $v_{lat}$ is the lateral velocity. These metrics collectively capture abrupt changes in acceleration and steering, providing a comprehensive measure of passenger comfort and driving stability.

### A.7  More Qualitative Results

Fig. 7 presents additional qualitative comparisons across various driving scenarios, including detours, crawling in dense traffic, traffic congestion, and U-turn maneuvers. The results highlight the effectiveness of our approach in generating smoother trajectories, enhancing collision avoidance, and improving adaptability in complex environments.

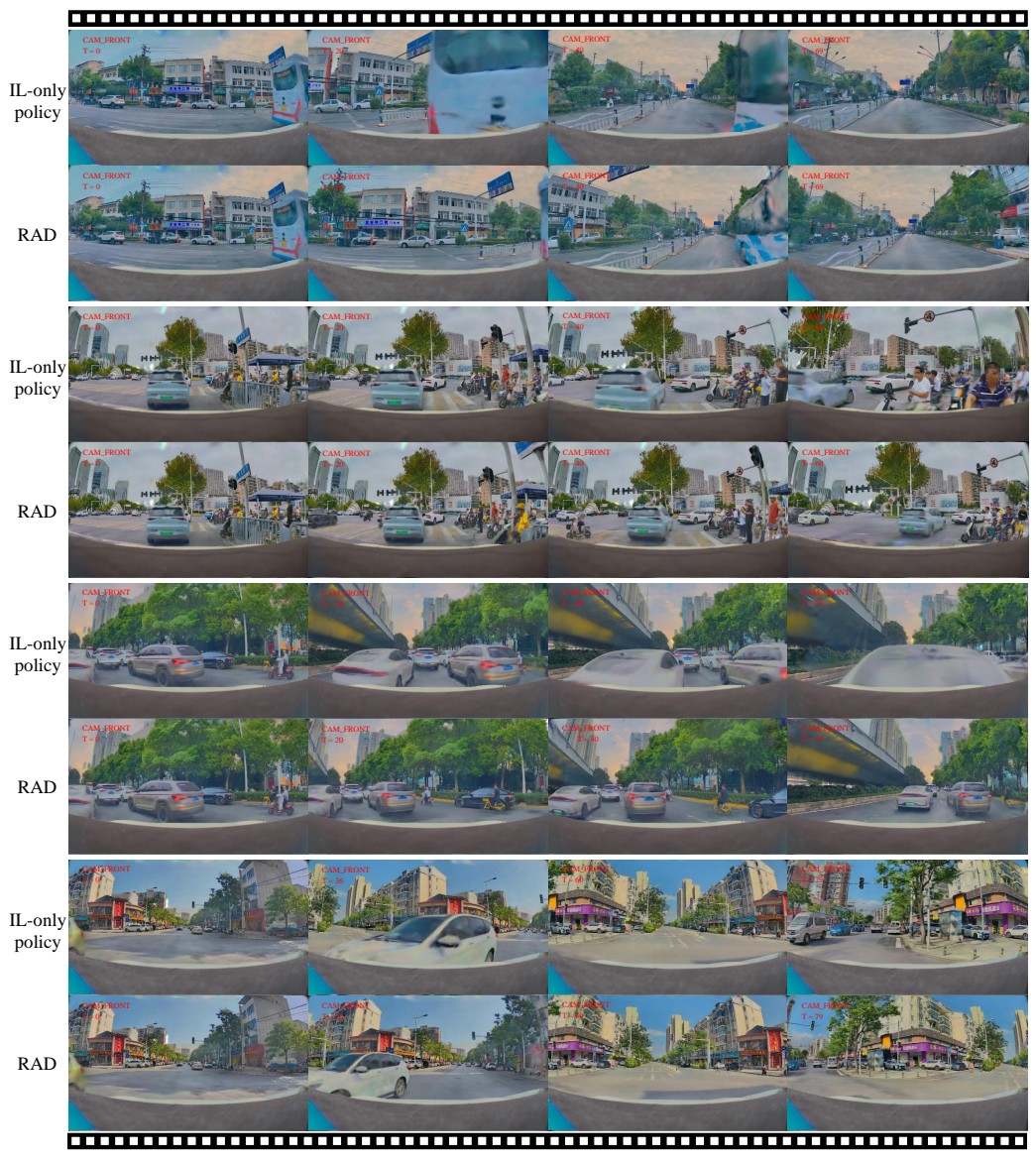

Figure 7: **More Qualitative Results.** Comparison between the IL-only policy and RAD in various driving scenarios: Detour (Rows 1-2), Crawl in Dense Traffic (Rows 3-4), Traffic Congestion (Rows 5-6), and U-turn(Rows 7-8).

