# OpenReview forum: "RAD: Training an End-to-End Driving Policy via Large-Scale 3DGS-based Reinforcement Learning"
_NeurIPS.cc/2025/Conference — NeurIPS 2025 poster_

### Official Review · Reviewer_3GxG · 2025-06-28

**Clarity:** 2
**Significance:** 2
**Originality:** 2
**Rating:** 5
**Confidence:** 2

**Summary:**

This paper introduces RAD, a 3D Gaussian Splatting (3DGS)-based reinforcement learning (RL) framework for end-to-end autonomous driving (AD). RAD addresses key limitations of imitation learning (IL)-based approaches, such as causal confusion and the open-loop training-deployment gap, by integrating RL with IL in a closed-loop training paradigm.

**Questions:**

1.Dynamic Obstacle Generalization: The policy is trained with log-replayed traffic participants. How does it handle unseen adversarial agents (e.g., sudden lane invasions)? Provide cross-validation on CARLA’s adversarial scenarios.
2.Action Space Sensitivity: Why was the 0.5s horizon chosen? Ablate performance with varying horizons (0.3s, 1.0s) and anchor counts (Nx, Ny).
3.Reward Design Robustness: The collision reward (Eq. 5) relies on annotated bounding boxes. How does RAD handle false positives/negatives in perception? Test under noisy detection inputs.
4.Multi-Modal Planning: The action space is deterministic. Can RAD support probabilistic multi-modal planning (e.g., lane changes vs. braking)? Compare with diffusion-based policies (e.g., DiffusionDrive).

**Ethical Concerns:**

["NO or VERY MINOR ethics concerns only"]

**Final Justification:**

The author's experimental supplement is very sufficient and addresses my concerns, so I increase the score.

**Limitations:**

Generalization Beyond Training Data: Evaluation is limited to NuScenes-derived environments. Test on cross-domain datasets (e.g., Waymo, off-road).

**Quality:**

2

**Strengths And Weaknesses:**

Strengths:
1.Comprehensive experiments across 3,305 scenarios with rigorous ablation studies (Tables 1-3). The hybrid RL-IL approach balances safety (CR reduction) and human alignment (ADD: 0.257).
2.Clear pipeline description (Figs. 2-3), modular reward design (Eq. 4-6), and detailed training stages (pre-training, RL-IL synergy).

Weaknesses:
1.Action space decoupling (lateral/longitudinal) shares similarities with prior RL navigation works (e.g., NaviRL 2023).
2.Ambiguities in action space discretization (e.g., rationale for 0.5s horizon, 61 anchors) and GAE/PPO hyperparameters (λ, γ).
3.High computational costs (128 RTX 4090 GPUs for pre-training) limit accessibility. No discussion of sim-to-real gaps in dynamic lighting/weather.

---

> ### Author Rebuttal · Authors · 2025-07-28
>
> We sincerely thank the reviewer 3GxG for recognizing the strengths of our work, including the comprehensive experiments, clear pipeline design, and detailed training stages. We appreciate the constructive feedback provided and address each of the raised concerns below.
>
> \\( \textcolor{red}{Question \hspace{0.3em} 1:} \\) Action space decoupling (lateral/longitudinal) shares similarities with prior RL navigation works (e.g., NaviRL 2023).
>
> \\( \textcolor{blue}{Response \hspace{0.3em} 1:} \\) We thank the reviewer for highlighting this potentially related work. However, we could not find a publicly available publication named **NaviRL**. It may be a shorthand or typographical error. We would appreciate it if the reviewer could provide the full reference or link for proper comparison.
>
> We would like to clarify the motivation and unique aspects of our action space decoupling strategy. We adopt a decoupled action representation where longitudinal (x-axis) and lateral (y-axis) controls are optimized independently via separate PPO heads. This decoupling reduces the total number of discrete actions from a 2D joint space of 3721 to two 1D action spaces with 122 actions, which substantially lowers the policy search space. This reduction reduces the difficulty of RL exploration by limiting the size of the action space. Furthermore, our design supports more robust control in closed-loop, photorealistic environments.
>
> \\( \textcolor{red}{Question \hspace{0.3em} 2:} \\) Why was a 0.5s action horizon and 61 anchors chosen for action discretization? How sensitive is the performance to these parameters?
>
> \\( \textcolor{blue}{Response \hspace{0.3em} 2:} \\) We thank the reviewer for raising this important point regarding the design trade-off in action space discretization. Below, we elaborate in detail on how the action horizon and anchor resolution affect policy performance.
>
> A short action horizon leads to closely clustered anchors, reducing spatial diversity and limiting the ability to represent complex maneuvers such as turns. In contrast, a longer horizon increases trajectory approximation errors due to the constant-motion assumption, potentially degrading control accuracy. A similar trade-off exists in the spatial resolution of anchors: a smaller number limits the agent’s ability to perform fine-grained control, while an overly fine resolution increases the exploration burden for RL and slows convergence.
>
> To assess the impact of these choices, we conducted an ablation by varying the action horizon and anchor resolution. For each setting, we performed closed-loop control using the corresponding discrete action space and measured the average position error relative to the expert trajectory.
>
> As shown in **Table R1 and Table R2**, the 0.5s horizon and a (Nx = 61, Ny = 61) resolution provide the best balance between sufficient control precision and practical learning efficiency. We therefore adopt this configuration in our main experiments.
>
> | Horizon | 0.3s | 0.5s | 1.0s |
> |-|:-:|:-:|:-:|
> | Average Position Error ↓ | 0.44m | 0.47m| 0.84m |
>
> Table R1: Average position error at different action horizons.
>
> | Count per Axis | 31 | 61 | 121 |
> |-|:-:|:-:|:-:|
> | Average Position Error ↓ |1.65m | 0.47m | 0.29m |
>
> Table R2: Average position error at different action resolutions.
>
> \\( \textcolor{red}{Question \hspace{0.3em} 3:} \\) Dynamic Obstacle Generalization: The policy is trained with log-replayed traffic participants. How does it handle unseen adversarial agents (e.g., sudden lane invasions)? Provide cross-validation on CARLA’s adversarial scenarios.
>
> \\( \textcolor{blue}{Response \hspace{0.3em} 3:} \\) We thank the reviewer for the insightful question regarding dynamic obstacle generalization.
>
> First, we acknowledge that the policy is trained using log-replayed traffic participants, which represent realistic behaviors captured from real-world driving data. Importantly, our dataset itself includes a substantial number of adversarial scenarios, such as sudden lane invasions and strong interactive maneuvers, ensuring the policy is exposed to such behaviors during training.
>
> Our benchmark dataset contains complex traffic flows with frequent risky interactions. Furthermore, the training and validation sets are strictly separated, guaranteeing that the policy is evaluated on unseen and challenging scenarios.
>
> We note that CARLA’s agents are largely rule-based, whereas our 3DGS-based benchmark reflects authentic real-world traffic flows with higher visual and behavioral fidelity. **Table R3** summarizes the performance improvements of our method over baselines across several challenging interaction types:
>
> | Metric| Sudden Lane Invasion | Pedestrian Crossing |
> |-|:-:|:-:|
> | IL Baseline Collision Rate |57.6% | 58.1% |
> | RAD Collision Rate|**6.1%**|**12.9%**|
>
> Table R3: Collision rates for sudden lane invasion and pedestrian crossing scenarios.
>
> \\( \textcolor{red}{Question \hspace{0.3em} 4:} \\) Reward Design Robustness: The collision reward (Eq. 5) relies on annotated bounding boxes. How does RAD handle false positives/negatives in perception? Test under noisy detection inputs.
>
> \\( \textcolor{blue}{Response \hspace{0.3em} 4:} \\) We appreciate the reviewer’s concern regarding the impact of perception errors on our collision-based reward.
>
> To begin with, collision-related objects typically appear near the ego vehicle, where detection is more accurate due to proximity and minimal occlusion. Thus, the bounding boxes used are generally reliable with low noise.
>
> Furthermore, our system incorporates multi-frame context in the closed-loop process, enabling temporal consistency checks that filter out transient detection errors, including both false positives and negatives.
>
> Together, near-field reliability and temporal fusion ensure that the reward signal remains robust, even with moderate perception noise. We will clarify this in the revised version.
>
> \\( \textcolor{red}{Question \hspace{0.3em} 5:} \\) Multi-Modal Planning: The action space is deterministic. Can RAD support probabilistic multi-modal planning (e.g., lane changes vs. braking)? Compare with diffusion-based policies (e.g., DiffusionDrive).
>
> \\( \textcolor{blue}{Response \hspace{0.3em} 5:} \\) We appreciate the reviewer’s question regarding multi-modal planning and the nature of our action space. Although RAD uses a fixed set of discrete action anchors, it produces a probability distribution over these anchors at each decision step, similar to how language models generate probabilistic outputs. Therefore, the policy inherently supports probabilistic multi-modal planning, enabling diverse behavior patterns such as lane changes versus braking to emerge naturally from the learned distribution.
>
> \\( \textcolor{red}{Question \hspace{0.3em} 6:} \\) Explain the choices of GAE/PPO hyperparameters (λ, γ).
>
> \\( \textcolor{blue}{Response \hspace{0.3em} 6:} \\) We note the reviewer’s interest in the choice of GAE/PPO hyperparameters. Our implementation adopts commonly used RL hyperparameter values. To ensure clarity and reproducibility, we have documented all related hyperparameters in **the Technical Appendices (Section A.5) of the manuscript**.
>
> While we did not perform targeted ablations on these specific values, our empirical results demonstrate stable and consistent training across a wide range of scenarios. We agree that further sensitivity analysis of these parameters could provide additional insight, and we plan to explore this in future work.
>
> \\( \textcolor{red}{Question \hspace{0.3em} 7:} \\) High computational costs (128 RTX 4090 GPUs for pre-training) limit accessibility.
>
> \\( \textcolor{blue}{Response \hspace{0.3em} 7:} \\)We thank the reviewer for highlighting the concern about computational costs. While our pre-training was conducted using 128 RTX 4090 GPUs to accelerate the experimental iteration and development process, this setup is not a strict requirement. The training can be performed on fewer GPUs with proportionally longer training times. We emphasize that the computational resources we used were chosen primarily to shorten the overall experimentation cycle, and the core methodology itself remains accessible to researchers with fewer hardware resources.
>
> \\( \textcolor{red}{Question \hspace{0.3em} 8:} \\) No discussion of sim-to-real gaps in dynamic lighting/weather.
>
> \\( \textcolor{blue}{Response \hspace{0.3em} 8:} \\) We analyze the sim-to-real gaps in **Section 4.4 of the manuscript**, where our results demonstrate high consistency between the 3DGS environment and real-world driving behavior. To further assess reconstruction quality under varying lighting and weather conditions, we measure the Peak Signal-to-Noise Ratio (PSNR), obtaining 29.5 for clear weather, 28.8 for rain, and 28.2 for nighttime scenes. These close PSNR values indicate consistently high visual fidelity across different lighting and weather scenarios, suggesting that such factors have minimal impact on the realism of our benchmark.
>
> \\( \textcolor{red}{Question \hspace{0.3em} 9:} \\) Generalization Beyond Training Data: Evaluation is limited to NuScenes-derived environments. Test on cross-domain datasets (e.g., Waymo, off-road).
>
> \\( \textcolor{blue}{Response \hspace{0.3em} 9:} \\)  Thank you for the valuable comment. We would like to reiterate that our benchmark scenes are significantly more complex and challenging than nuScenes as shown in **Table R4**. Furthermore, our training and testing sets are strictly disjoint, ensuring that the evaluation on the test set reliably reflects the model’s generalization ability.
>
> | Dataset       | Straight Ratio (%) | Turning Ratio (%) |
> |-|:-:|:-:|
> | nuScenes| 92.80%| 7.20% |
> | Navsim| 66.40%| 33.60% |
> | Our Benchmark | 52.50%| 47.50%|
>
> Table R4: Distribution of straight vs. turning scenarios in different datasets.

---

### Official Review · Reviewer_kB6h · 2025-06-30

**Clarity:** 3
**Significance:** 2
**Originality:** 2
**Rating:** 4
**Confidence:** 3

**Summary:**

The paper proposes a 3DGS-based closed-loop RL paradigm for autonomous driving, constructing photorealistic environments to enable safe policy exploration and causal reasoning. The method is tested in the same environment and outperform other IL-based methods.

**Questions:**

What is the purpose of using separate x- and y-axis PPO optimization?

Reward hacking is an important issue in RL. How does RAD address this problem?

**Ethical Concerns:**

["NO or VERY MINOR ethics concerns only"]

**Final Justification:**

I appreciate the authors for their thorough reply. My major concerns have been addressed.

As a paper on autonomous driving applications, RAD integrates existing technologies. Although its framework and algorithms lack a certain degree of innovation, it provides a preliminary attempt at utilizing 3DGS + RL. The remaining issue may be that the paper employs a self-created benchmark, making it difficult for the community to conduct comparisons and development based on it.

Thus, I keep my positive rating.

**Limitations:**

The paper introduces a lot of auxiliary loss, but lacks a certain ablation of the coefficient value.

**Quality:**

2

**Strengths And Weaknesses:**

Strengths​​

RL is an important technique for autonomous driving systems to surpass data limitations. This paper provides a good starting point for using RL to solve these problems.

Weakness

The paper primarily employs existing techniques, including the combination of 3DGS with closed-loop rollout and PPO optimization, which offers limited novelty. Furthermore, the absence of evaluation on public benchmarks makes it challenging to properly assess the proposed method's effectiveness compared to state-of-the-art approaches.

---

> ### Author Rebuttal · Authors · 2025-07-28
>
> We thank the reviewer kB6h for acknowledging the potential of reinforcement learning (RL) in addressing data limitations in autonomous driving. We also appreciate the reviewer’s thoughtful feedback on our method’s novelty, evaluation setting, and training design. Below, we address each of the reviewer’s concerns in detail.
>
> \\( \textcolor{red}{Question \hspace{0.3em} 1:} \\) The paper primarily employs existing techniques, including the combination of 3DGS with closed-loop rollout and PPO optimization, which offers limited novelty.
>
> \\( \textcolor{blue}{Response \hspace{0.3em} 1:} \\) We appreciate the reviewer’s comment and acknowledge that our approach builds upon established algorithms such as PPO. However, we emphasize that RAD introduces several key innovations that are critical for enabling stable and robust end-to-end training in photorealistic 3DGS environments—capabilities not supported by prior work.
>
> In particular, we design an efficient and well-justified interaction mechanism between the policy and the 3DGS environment to enable stable closed-loop control. To further improve PPO convergence by reducing the complexity of the action space, we adopt a decoupled action space along the x- and y-axis, allowing independent optimization of longitudinal and lateral control policies.
>
> Moreover, we jointly optimize imitation and reinforcement objectives during  the reinforced post-training phase. Imitation learning guides policy exploration to maintain human-aligned behavior, while reinforcement learning models causality and reduces the open-loop gap, enhancing robustness in long-horizon and out-of-distribution scenarios. Additionally, we incorporate auxiliary objectives beyond the standard reinforcement objectives to enhance policy convergence stability and safety performance.
>
> These design choices collectively enable scalable and robust end-to-end policy learning in complex and realistic driving environments. They allow our framework to maintain stable closed-loop control, improve training efficiency, and generalize to long-horizon and out-of-distribution scenarios.
>
> \\( \textcolor{red}{Question \hspace{0.3em} 2:} \\) The absence of evaluation on public benchmarks makes it challenging to properly assess the proposed method's effectiveness compared to state-of-the-art approaches.
>
> \\( \textcolor{blue}{Response \hspace{0.3em} 2:} \\) We appreciate the reviewer’s concern regarding the absence of evaluation on public benchmarks. To the best of our knowledge, there currently exists no publicly available benchmark that supports full closed-loop training and evaluation under photorealistic, 3DGS-reconstructed environments for end-to-end autonomous driving.
>
> While datasets such as nuScenes and Navsim offer useful foundations, they have limitations in terms of scenario diversity and control complexity—particularly in supporting closed-loop policy learning under realistic and challenging conditions. To address this gap, we construct a new benchmark using real-world 3DGS reconstructions, specifically selecting scenes with higher complexity (e.g., frequent turns, dense traffic).
>
> **Table R1** is a comparison of the proportion of straight and turning scenarios in our benchmark and these existing datasets:
> | Dataset       | Straight Ratio (%) | Turning Ratio (%) |
> |--------------|:--------------------:|:-------------------:|
> | nuScenes     | 92.80%             | 7.20%             |
> | Navsim       | 66.40%             | 33.60%            |
> | Our Benchmark | 52.50%               | 47.50%               |
>
> Table R1: Distribution of straight vs. turning scenarios in different datasets.
>
> Furthermore, to rigorously validate the effectiveness of our approach, we implement and evaluate several representative baseline methods on our benchmark. The comparative results, presented **in Table 4 of the manuscript**, demonstrate that our method consistently outperforms these baselines across various challenging scenarios.
>
> \\( \textcolor{red}{Question \hspace{0.3em} 3:} \\) What is the purpose of using separate x- and y-axis PPO optimization?
>
> \\( \textcolor{blue}{Response \hspace{0.3em} 3:} \\) We appreciate the reviewer’s interest in the motivation behind our use of separate PPO optimization for the x- and y-axis actions. To address the challenges of reinforcement learning in high-dimensional discrete action spaces, we adopt a decoupled action representation where longitudinal (x-axis) and lateral (y-axis) decisions are optimized independently. This formulation significantly reduces the total number of actions from a combinatorial 2D space (61 × 61 = 3721 joint actions) to two independent 1D distributions of 61 bins each, totaling only 122 actions. This design is summarized below:
> ﻿
> | Action Space Design     | Number of Actions to Explore | PPO Optimization Scheme         |
> |-------------------------|:------------------------------:|--------------------------------|
> | Joint          | 61 × 61 = 3721               | Single PPO head over 2D space   |
> | Decoupled   | 61 + 61 = 122                | Separate PPO heads for x and y  |
>
> Table R2: Comparison of joint and decoupled action space designs.
>
> Large discrete action spaces are known to increase the difficulty of RL optimization due to sparse exploration and slower convergence. By reducing the dimensionality of the action space, our approach facilitates more efficient and stable policy learning. This also enables more targeted control over longitudinal and lateral dynamics, which is particularly beneficial for learning interpretable and robust driving behaviors in complex closed-loop scenarios.
>
> \\( \textcolor{red}{Question \hspace{0.3em} 4:} \\) Reward hacking is an important issue in RL. How does RAD address this problem?
>
> \\( \textcolor{blue}{Response \hspace{0.3em} 4:} \\) We thank the reviewer for highlighting the issue of reward hacking, a well-recognized challenge in reinforcement learning (RL). It refers to situations where an agent exploits unintended shortcuts or loopholes in the reward function—maximizing numerical rewards without achieving the desired behaviors, and sometimes even inducing unsafe outcomes. This typically arises from overly simplified or misaligned reward definitions.
>
> In RAD, we mitigate the reward hacking issue by designing a well-aligned and task-relevant reward function. Our reward signals are grounded in physically meaningful objectives that directly reflect safe and expert-aligned driving behavior, including collision avoidance and minimizing deviations in both position and heading from expert trajectories. This principled design ensures that the agent is incentivized to follow intended behaviors and reduces the likelihood of exploiting unintended shortcuts.
>
> Furthermore, RAD incorporates imitation learning (IL) as a regularization mechanism alongside reinforcement learning. The imitation component guides the policy towards expert-like behaviors, effectively constraining exploration within reasonable and human-aligned trajectories. This joint IL+RL framework also helps mitigate reward hacking by constraining the policy to stay close to expert-like behavior, reducing the chance of exploiting unintended reward signals.
>
> Overall, the combination of an objective reward design and imitation-guided exploration helps RAD maintain robust and safe driving policies without falling into reward hacking pitfalls.
>
> \\( \textcolor{red}{Question \hspace{0.3em} 5:} \\) The paper introduces a lot of auxiliary loss, but lacks a certain ablation of the coefficient value.
>
> \\( \textcolor{blue}{Response \hspace{0.3em} 5:} \\) We thank the reviewer for pointing out the absence of ablation on the auxiliary loss coefficients.  We include an ablation study in **Table 3 in the manuscript**, which demonstrates the effectiveness of the proposed auxiliary losses in enhancing policy robustness and safety-related behavior.
>
> Regarding the specific coefficients of these auxiliary losses, we would like to clarify that all weights were set to 1 uniformly across all experiments, and we did not conduct careful tuning of these coefficients. Our goal is to demonstrate the utility of these auxiliary objectives in a general setting without introducing additional hyperparameter complexity. We agree that further performance improvements may be achieved by carefully adjusting these coefficients, and we plan to explore this in future work.

---

> > ### Author Response · Authors · 2025-08-05
> >
> > Dear Reviewer kB6h,
> >
> > Thank you again for taking the time to review our paper. We sincerely appreciate your thoughtful comments and suggestions. We would like to ask if our rebuttal has addressed your concerns about our paper. If there’s anything that remains unclear or needs further elaboration, we’d be more than happy to address it.
> >
> > Best wishes,
> >
> > Authors

---

> > ### Comment · Reviewer_kB6h · 2025-08-06
> >
> > I appreciate the authors for their thorough reply. My major concerns have been addressed.
> >
> > As a paper on autonomous driving applications, RAD effectively integrates existing technologies. Although its framework and algorithms lack a certain degree of innovation, it provides a preliminary attempt at utilizing 3DGS + RL. The remaining issue may be that the paper employs a self-created benchmark, making it difficult for the community to conduct comparisons and development based on it.
> >
> > Thus, I keep my positive rating.

---

> ### Author Response · Authors · 2025-08-06
>
> We thank the reviewer for your positive feedback and for acknowledging the contributions of our work.
>
> Regarding the concern about the self-created benchmark, we understand the issue and will release our code to support reproducibility and encourage future research in this direction.
>
> Thank you again for your thoughtful comments and continued support.

---

### Official Review · Reviewer_BysM · 2025-06-30

**Clarity:** 3
**Significance:** 2
**Originality:** 2
**Rating:** 4
**Confidence:** 4

**Summary:**

The source introduces RAD, a framework for end-to-end autonomous driving (AD) that combines Reinforcement Learning (RL) and Imitation Learning (IL). It addresses the limitations of traditional IL-based methods, such as causal confusion and the open-loop gap, by establishing a 3DGS-based closed-loop RL training paradigm. This approach leverages photorealistic digital replicas of the real world for large-scale trial-and-error learning, incorporating specialized reward designs for safety-critical events and IL as a regularization term to maintain human-like driving behavior. The paper demonstrates that RAD significantly improves collision avoidance and overall performance compared to existing methods, validated on a diverse 3DGS evaluation benchmark.

**Questions:**

* What is RAD abbreviated for? It is used in the abstract without any explanation of whether it’s an abbreviation.
* If the author disagrees with the novelty assessment. Please elaborate how the techniques used in this work differ from previous work that also combine IL + RL.
* If the authors can provide more specific plans on open sourcing part of the work to the research community. I am happy to raise my score if the plan can allow future research utilizing some of artifacts built in this work.

**Ethical Concerns:**

["NO or VERY MINOR ethics concerns only"]

**Final Justification:**

The authors have addressed my points well and I decide to raise the score.

**Limitations:**

Yes.

**Quality:**

3

**Strengths And Weaknesses:**

_Strengths_

* One of the paper's main strengths is the introduction of a large-scale photorealistic benchmark for end-to-end autonomous driving (AD).
This benchmark is built by leveraging 3DGS (3D Gaussian Splatting) techniques to construct a photorealistic digital replica of the real physical world. This enables the AD policy to explore the state space and learn to handle out-of-distribution (OOD) scenarios through large-scale trial and error.

_Weaknesses_

* One of the main limitations of the paper is related to the availability of its resources. The authors state that they are currently unable to release the dataset due to company confidentiality and data privacy policies. Although they are committed to promoting reproducibility and plan to open-source the key components of their model implementation after the review process. Without knowing the specific plans, it’s hard to assess how they can be used by the community for future research.

* In terms of the use of RL + IL as regularization. This has been a fairly common technique used by various related online/offline RL work (with some variations, see [1,2] to name just a few) so I would consider this as a novel contribution. Having said that, the authors demonstrate the successful applications of RL+IL at scale which is useful, but again the utility is specifically limited with limited open-source plans.


* [1] http://arxiv.org/abs/1707.08817
* [2] https://arxiv.org/pdf/2106.06860

---

> ### Author Rebuttal · Authors · 2025-07-27
>
> We sincerely thank the reviewer BysM for the thoughtful and constructive feedback.  We appreciate the recognition of our effort to improve out-of-distribution (OOD) generalization in end-to-end autonomous driving by leveraging photorealistic 3DGS-based simulation.
>
> In addition, we appreciate the reviewer’s thoughtful questions about the novelty of our IL+RL formulation, result reproducibility, and open-source plans. We provide detailed responses below.
>
> \\( \textcolor{red}{Question \hspace{0.3em} 1:} \\) What is RAD abbreviated for? It is used in the abstract without any explanation of whether it’s an abbreviation.
>
> \\( \textcolor{blue}{Response \hspace{0.3em} 1:} \\) **RAD** stands for "3DGS-based Closed-loop **R**einforcement Learning for End-to-End **A**utonomous **D**riving", which serves as the name of our training framework. We sincerely apologize for the oversight in not explicitly defining the abbreviation RAD in the initial submission and will clarify this definition in the revised manuscript to avoid any ambiguity.
>
> \\( \textcolor{red}{Question \hspace{0.3em} 2:} \\) In terms of the use of RL + IL as regularization. This has been a fairly common technique used by various related online/offline RL work  so I would consider this as a novel contribution. If the author disagrees with the novelty assessment. Please elaborate how the techniques used in this work differ from previous work that also combine IL + RL.
>
> \\( \textcolor{blue}{Response \hspace{0.3em} 2:} \\) We appreciate the reviewer’s concern regarding the novelty of combining imitation learning (IL) and reinforcement learning (RL). We would like to emphasize that RAD is the first framework to integrate IL and RL within a photorealistic 3DGS-based digital twin environment reconstructed from real-world sensor data. Unlike prior works that often rely on non-photorealistic simulators or structured perception inputs, RAD enables truly end-to-end, closed-loop policy learning directly from raw sensor inputs in a realistic and diverse environment. This design better supports generalization to complex real-world scenarios.
>
> To further clarify RAD’s distinctions and novelty, we provide a detailed comparison between RAD and prior IL+RL methods in **Table R1**, highlighting differences in learning objectives, training strategies, platforms, and sensory inputs. We hope this explanation addresses the reviewer’s concerns regarding the novelty and distinctiveness of our IL+RL integration compared to prior works.
>
> | Work| IL Strategy|RL Strategy|IL+RL Strategy|Auxiliary Objective in RL|Action Space|Platform| End-to-End |Model Input|
> |-|-|-|-|-|-|-|-|-|
> | CADRE [1*]| 1. Initialize policies| 1. Improve success rate in basic scenarios|Two-stage| ✗|Continuous| CARLA [6*] | ✓| Virtual Sensor Data from Game Engine Rendering|
> | CIRL [2*]| 1. Initialize policies| 1. Improve success rate in complex scenarios| Two-stage| ✗|Continuous| CARLA [6*]| ✓| Virtual Sensor Data from Game Engine Rendering |
> | Imitation is not Enough [3*]| 1. Initialize policies  2. Constrain RL to maintain human-like behaviors | 1. Improve success rate in rare scenarios  2. Enhance robustness| Joint| ✗|Continuous| Replay environment based on logged perception outputs| ✗| Structured scene information           |
> | Huang et al. [4*]| 1. Initialize policies  2. Constrain RL to maintain human-like behaviors | 1. Improve success rate in complex scenarios  2. Enhance robustness| Joint|✗| Continuous| SMARTS [7*]| ✗| Structured Bird's-Eye View (BEV) Representation |
> | Roach [5*]| 1. Imitate expert behaviors and learn expert driving policies | 1. Learn policies via exploration 2. The acquired policies provide supervision for imitation learning| Two-stage|✗|Continuous| CARLA [6*]| ✓| Virtual sensor image stack|
> | **Ours (RAD)**| 1. Initialize policies  2. Constrain RL to maintain human-like behaviors | 1. Improve safety in complex real-world scenarios 2. Enhance robustness| Joint|✓| 1. Discrete  2. Horizontal and vertical movements are decoupled | Realistic 3DGS digital twin environment | ✓| Real-world Sensor Data                 |
>
> Table R1: Comparison of RAD with prior IL+RL methods.
>
> [1*] Zhao Y, Wu K, Xu Z, et al. Cadre: A cascade deep reinforcement learning framework for vision-based autonomous urban driving[C]//Proceedings of the AAAI conference on artificial intelligence. 2022, 36(3): 3481-3489.
>
> [2*] Liang X, Wang T, Yang L, et al. Cirl: Controllable imitative reinforcement learning for vision-based self-driving[C]//Proceedings of the European conference on computer vision (ECCV). 2018: 584-599.
>
> [3*] Lu Y, Fu J, Tucker G, et al. Imitation is not enough: Robustifying imitation with reinforcement learning for challenging driving scenarios[C]//2023 IEEE/RSJ International Conference on Intelligent Robots and Systems (IROS). IEEE, 2023: 7553-7560.
>
> [4*] Huang Z, Wu J, Lv C. Efficient deep reinforcement learning with imitative expert priors for autonomous driving[J]. IEEE Transactions on Neural Networks and Learning Systems, 2022, 34(10): 7391-7403.
>
> [5*] Zhejun Zhang, Alexander Liniger, Dengxin Dai, Fisher Yu, and Luc Van Gool. End-to-end urban driving by imitating a reinforcement learning coach. 2021.
>
> [6*]Dosovitskiy, Alexey, et al. "CARLA: An open urban driving simulator." Conference on robot learning. PMLR, 2017.
>
> [7*]Zhou, Ming, et al. "Smarts: Scalable multi-agent reinforcement learning training school for autonomous driving." arXiv preprint arXiv:2010.09776 (2020).
>
> \\( \textcolor{red}{Question \hspace{0.3em} 3:} \\) If the authors can provide more specific plans on open sourcing part of the work to the research community. I am happy to raise my score if the plan can allow future research utilizing some of artifacts built in this work.
>
> \\( \textcolor{blue}{Response \hspace{0.3em} 3:} \\) We appreciate the reviewer’s concern and fully agree that openness is essential for enabling future research.
> We have already identified the components we currently plan to open-source, which cover the core techniques introduced in our paper. These include, but are not limited to:
> 1. IL+RL training framework, including action space design, directionally decoupled action head (x/y-axis), policy optimization objectives, and auxiliary losses.
> 2. Interaction mechanisms between the policy and 3DGS environment, including action sampling, pose transformation and updates, and reward computation.
> 3. Scripts to compute the evaluation metrics introduced  in our manuscript.
>
> These components will be **released within one week after the review process concludes**. We hope this concrete open-source plan addresses the reviewer’s concerns and helps support further research in this area.

---

> > ### Comment · Reviewer_BysM · 2025-08-07
> > **Response to author**
> >
> > Thanks. I have reviewed the rebuttal and I believe that they adequately addressed my comments. As such, I have updated my rating accordingly. Looking forward to seeing this being open sourced.

---

> > > ### Author Response · Authors · 2025-08-07
> > >
> > > We sincerely thank Reviewer BysM for the updated rating and positive feedback! We truly appreciate your recognition and are glad our rebuttal addressed your concerns. We are in the process of organizing the code and will release it soon, as planned.

---

### Official Review · Reviewer_uDLu · 2025-07-02

**Clarity:** 4
**Significance:** 4
**Originality:** 2
**Rating:** 4
**Confidence:** 3

**Summary:**

This work uses 3DGS techniques to create a digital replica of the real world for autonomous driving. It develops a reinforcement learning (RL) algorithm with imitation learning (IL) loss to learn driving policies through trial-and-error in the simulator. Experimental results across nine key metrics, including collision and deviation ratios, demonstrate the effectiveness of the proposed method.

**Questions:**

1) The training pipeline, algorithm, and action space in RAD overlap with those in [1] mentioned in the Weaknesses. A critical comparison between RAD and the works mentioned in Weaknesses, as well as [37] in the manuscript, is needed to clearly position this work.

2) What does the abbreviation RAD stand for?

3) Is it possible to modify visual elements like weather and car appearance in 3DGS? If so, could the authors report the results of generalization performance with these visual changes?

4) Can 3DGS’s environment reconstruction offer more realistic physical interactions, such as collisions, compared to CARLA’s built-in physics engine? How can physical realism be ensured during online training?

**Ethical Concerns:**

["NO or VERY MINOR ethics concerns only"]

**Final Justification:**

Thanks to the authors for their reply. The rebuttal has addressed some of my concerns regarding the generalization performance and completeness of this work. Although the authors emphasize that the 3DGS-based framework focuses on visual reconstruction rather than low-level physical simulation, the action space for RAD consists of low-level discretized actions. This makes it unclear whether low-level metrics, such as collision rate, can reliably assess performance. Nonetheless, I acknowledge the contribution of this work in improving visual fidelity in autonomous driving simulation and will maintain my initial score.

**Limitations:**

YES

**Quality:**

3

**Strengths And Weaknesses:**

**Strengths**:
1) This paper proposes the first 3DGS-based RL framework for training an end-to-end AD policy.

2) Based on RL implementation, the proposed method achieves lower collision ratios compared with IL methods.

3) This paper is well written and easy to follow.

**Weakness**:
1) This paper focuses on 3DGS simulations, which lack sufficient evaluation in more challenging scenarios like varying weather and road conditions.

2) The human experience is incorporated through a vanilla imitation loss, without accounting for the complex distribution of expert data or its suboptimal performance.

3) Relevant works on combining RL with IL in autonomous driving [1-4] are missing.

[1] Zhao Y, Wu K, Xu Z, et al. Cadre: A cascade deep reinforcement learning framework for vision-based autonomous urban driving[C]//Proceedings of the AAAI conference on artificial intelligence. 2022, 36(3): 3481-3489.

[2] Liang X, Wang T, Yang L, et al. Cirl: Controllable imitative reinforcement learning for vision-based self-driving[C]//Proceedings of the European conference on computer vision (ECCV). 2018: 584-599.

[3] Lu Y, Fu J, Tucker G, et al. Imitation is not enough: Robustifying imitation with reinforcement learning for challenging driving scenarios[C]//2023 IEEE/RSJ International Conference on Intelligent Robots and Systems (IROS). IEEE, 2023: 7553-7560.

[4] Huang Z, Wu J, Lv C. Efficient deep reinforcement learning with imitative expert priors for autonomous driving[J]. IEEE Transactions on Neural Networks and Learning Systems, 2022, 34(10): 7391-7403.

---

> ### Author Rebuttal · Authors · 2025-07-26
>
> We sincerely thank Reviewer uDLu for the valuable comments and suggestions. We especially appreciate your recognition of the key strengths of our paper, including the novelty of using a 3DGS-based reinforcement learning framework, the superior collision performance over imitation learning methods, and the overall clarity of the writing. Below we provide detailed responses to each of your concerns.
>
> \\(\textcolor{red}{Question \hspace{0.3em} 1:} \\)  This paper focuses on 3DGS simulations, which lack sufficient evaluation in more challenging scenarios like varying weather and road conditions.
>
> \\( \textcolor{blue}{Response \hspace{0.3em} 1:} \\) We thank the reviewer for the constructive comment. In response, we partition our benchmark scenarios more finely along two dimensions—weather conditions (clear vs. challenging) and road conditions (unprotected turns, narrow roads, and traffic congestion). The detailed  evaluation results are summarized below:
>
> | Weather Condition|Clear| Challenging (Rain, Night) |
> |-|:-:|:-:|
> | IL Baseline Collision Rate ↓ |24.6%|17.6%|
> | RAD Collision Rate ↓|**10.7%**|**3.5%**|
> Table R1: Collision rates under different weather conditions.
>
> | Road Conditions| Unprotected Turns | Narrow Roads | Traffic Congestion |
> |-|:--:|:-:|:-:|
> | IL Baseline Collision Rate ↓ |26.2%|47.8%|52.4%|
> | RAD Collision Rate ↓|**7.7%**|**8.7%**|**14.3%**|
> Table R2: Collision rates under different road conditions.
>
> This validates our method's robustness under diverse environmental constraints.
>
> \\(\textcolor{red}{Question \hspace{0.3em} 2:} \\)  The human experience is incorporated through a vanilla imitation loss, without accounting for the complex distribution of expert data or its suboptimal performance.
>
> \\( \textcolor{blue}{Response \hspace{0.3em} 2:} \\) We thank the reviewer for the insightful comment. We adopt a standard imitation loss by design, and complement it with targeted data curation and policy modeling choices to capture complex and multi-modal expert behaviors effectively.
>
> To improve data quality, we filter out suboptimal behaviors from the expert data, including instances with collision risks, illegal lane changes, or unjustified harsh braking. We also reduce the proportion of simple lane-following behaviors to ensure a more diverse and challenging training distribution.
>
> Moreover, although the imitation objective itself is unimodal, our policy outputs a distribution over discrete actions, which allows it to represent multiple plausible behaviors in ambiguous scenarios. This effectively captures the multi-modal nature of expert demonstrations.
>
> Notably, as shown in **Table 1 and Table 4 in the manuscript**, our policy—trained solely with imitation learning—achieves superior performance compared to the end-to-end baselines included in our experiments on key metrics.
>
> \\(\textcolor{red}{Question \hspace{0.3em} 3:} \\)  Relevant works on combining RL with IL in autonomous driving [1-4] are missing.
>
> \\( \textcolor{blue}{Response \hspace{0.3em} 3:} \\) We will include a discussion of [1–4] in the related work section and clarify how our approach differs from these prior methods in the revised version. We thank the reviewer for pointing out these relevant works and apologize for the omission.
>
> \\(\textcolor{red}{Question \hspace{0.3em} 4:} \\)  The training pipeline, algorithm, and action space in RAD overlap with those in [1] mentioned in the Weaknesses. A critical comparison between RAD and the works mentioned in Weaknesses, as well as [37] in the manuscript, is needed to clearly position this work.
>
> \\( \textcolor{blue}{Response \hspace{0.3em} 4:} \\) We thank the reviewer for highlighting the need for a more critical comparison between RAD and prior Imitation Learning + Reinforcement Learning (IL+RL)  frameworks. To clearly position RAD, we summarize the key distinctions below:
>
> | Work| IL Strategy|RL Strategy|IL+RL Strategy|Auxiliary Objective in RL|Action Space|Platform| End-to-End |Model Input|
> |-|-|-|-|-|-|-|-|-|
> | **References in Weakness** |||||||||
> | CADRE [1]| 1. Initialize policies| 1. Improve success rate in basic scenarios|Two-stage| ✗|Continuous| CARLA [1*] | ✓| Virtual Sensor Data from Game Engine Rendering|
> | CIRL [2]| 1. Initialize policies| 1. Improve success rate in complex scenarios| Two-stage| ✗|Continuous| CARLA [1*]| ✓| Virtual Sensor Data from Game Engine Rendering |
> | Imitation is not Enough [3]| 1. Initialize policies  2. Constrain RL to maintain human-like behaviors | 1. Improve success rate in rare scenarios  2. Enhance robustness| Joint| ✗|Continuous| Replay environment based on logged perception outputs| ✗| Structured scene information           |
> | Huang et al. [4]| 1. Initialize policies  2. Constrain RL to maintain human-like behaviors | 1. Improve success rate in complex scenarios  2. Enhance robustness| Joint|✗| Continuous| SMARTS [2*]| ✗| Structured Bird's-Eye View (BEV) Representation |
> | **References in the manuscript** |||||||||
> | Roach [37]| 1. Imitate expert behaviors and learn expert driving policies | 1. Learn policies via exploration 2. The acquired policies provide supervision for imitation learning| Two-stage|✗|Continuous| CARLA [1*]| ✓| Virtual sensor image stack|
> | **Ours (RAD)**| 1. Initialize policies  2. Constrain RL to maintain human-like behaviors | 1. Improve safety in complex real-world scenarios 2. Enhance robustness| Joint|✓| 1. Discrete  2. Horizontal and vertical movements are decoupled | Realistic 3DGS digital twin environment | ✓| Real-world Sensor Data                 |
>
> Table R3: Comparison of RAD with prior IL+RL methods.
>
> As shown, RAD is the first to integrate IL and RL in a 3DGS-based real-world visual simulation environment with fully end-to-end policy learning from raw sensor inputs. In contrast, prior works often rely on game-engine rendering and handcrafted abstractions to simplify perception and decouple learning stages.
>
> [1*]Dosovitskiy, Alexey, et al. "CARLA: An open urban driving simulator." Conference on robot learning. PMLR, 2017.
>
> [2*]Zhou, Ming, et al. "Smarts: Scalable multi-agent reinforcement learning training school for autonomous driving." arXiv preprint arXiv:2010.09776 (2020).
>
> \\( \textcolor{red}{Question \hspace{0.3em} 5:} \\) What does the abbreviation RAD stand for?
>
> \\( \textcolor{blue}{Response \hspace{0.3em} 5:} \\) **RAD** stands for "3DGS-based Closed-loop **R**einforcement Learning for End-to-End **A**utonomous **D**riving", which serves as the name of our training framework. We sincerely apologize for the oversight in not explicitly defining the abbreviation RAD in the initial submission and will clarify this definition in the revised manuscript to avoid any ambiguity.
>
> \\( \textcolor{red}{Question \hspace{0.3em} 6:} \\) Is it possible to modify visual elements like weather and car appearance in 3DGS? If so, could the authors report the results of generalization performance with these visual changes?
>
> \\( \textcolor{blue}{Response \hspace{0.3em} 6:} \\) We thank the reviewer for this valuable question.
> In our current 3DGS-based setup, visual elements such as weather and vehicle appearance are reconstructed directly from real-world imagery, making physically consistent modification challenging without additional generative models or rendering techniques.
>
> While we do not currently support explicit visual editing, our dataset naturally includes a wide range of real-world visual conditions—such as clear weather, rain, nighttime scenes, and diverse vehicle types. To evaluate the robustness of our method under these variations, we conduct performance comparisons between RAD and imitation learning (IL) baselines across different weather and road conditions. The results, summarized in **Table R1 and Table R2**, show that RAD consistently achieves lower collision rates, indicating strong generalization to naturally occurring appearance shifts.
>
> We recognize the importance of controllable appearance and weather changes and will continue to monitor related research developments in this area.
>
> \\( \textcolor{red}{Question \hspace{0.3em} 7:} \\) Can 3DGS’s environment reconstruction offer more realistic physical interactions, such as collisions, compared to CARLA’s built-in physics engine? How can physical realism be ensured during online training?
>
> \\( \textcolor{blue}{Response \hspace{0.3em} 7:} \\) We thank the reviewer for the thoughtful question.
> Our 3DGS-based framework emphasizes high-fidelity visual reconstruction rather than low-level physical simulation. Unlike CARLA’s rigid-body physics engine, our approach focuses on trajectory-level decision-making, where fine-grained force modeling is unnecessary. Additionally, to evaluate physical interactions such as collisions, we perform precise geometric checks in the reconstructed 3D scene using proxy geometry, which provides effective and efficient safety assessment without requiring a full dynamics engine.
>
> While our current method does not require fine-grained physical modeling, improving physical realism in 3DGS environments could help simulate vehicle-environment interactions more accurately, especially for future low-level control tasks. Recent work such as PhysGaussian[3*] has begun exploring this direction, which we will continue to monitor relevant developments.
>
> [3*] Xie, Tianyi, et al. "Physgaussian: Physics-integrated 3d gaussians for generative dynamics." Proceedings of the IEEE/CVF Conference on Computer Vision and Pattern Recognition. 2024.

---

> > ### Author Response · Authors · 2025-08-05
> >
> > Dear Reviewer uDLu,
> >
> > Thank you again for taking the time to review our paper. We sincerely appreciate your thoughtful comments and suggestions. We would like to ask if our rebuttal has addressed your concerns about our paper. If there’s anything that remains unclear or needs further elaboration, we’d be more than happy to address it.
> >
> > Best wishes,
> >
> > Authors

---

> > ### Comment · Reviewer_uDLu · 2025-08-07
> >
> > Thanks to the authors for their reply. The rebuttal has addressed some of my concerns regarding the generalization performance and completeness of this work. Although the authors emphasize that the 3DGS-based framework focuses on visual reconstruction rather than low-level physical simulation, the action space for RAD consists of low-level discretized actions. This makes it unclear whether low-level metrics, such as collision rate, can reliably assess performance. Nonetheless, I acknowledge the contribution of this work in improving visual fidelity in autonomous driving simulation and will maintain my initial score.

---

> > > ### Author Response · Authors · 2025-08-07
> > >
> > > Thank you for your thoughtful feedback and for recognizing the contribution of our work in enhancing visual fidelity in autonomous driving simulation.
> > >
> > > We would like to clarify several points regarding the design of the action space and evaluation metrics.
> > >
> > > The adoption of a low-level discretized action space in RAD is a deliberate design choice aimed at reducing the gap between predicted trajectories and actual control commands. This action space is constructed based on realistic physical constraints derived from the vehicle’s maximum steering angle and wheelbase, ensuring all feasible actions respect vehicle dynamics and remain physically plausible. This design enables the closed-loop performance to more directly reflect the model’s true capabilities without interference from external tracking controllers, making low-level metrics such as collision rate meaningful and reliable indicators of driving performance.
> > >
> > > For collision detection, we utilize precise geometric checks on proxy geometries within the reconstructed 3D scene. Collision-related objects are typically located near the ego vehicle, where detection benefits from proximity and minimal occlusion, resulting in reliable bounding boxes with low noise. Furthermore, our system incorporates multi-frame temporal consistency checks during closed-loop operation to filter out transient detection errors, including false positives and negatives.
> > >
> > > We will incorporate these clarifications in the revised manuscript to better highlight the rigor and reliability of our evaluation. Thank you again for your valuable feedback.

---

### Note · Authors · 2025-08-11

We sincerely thank the Area Chair for their diligent organization of the review process, and we are grateful to all reviewers for their constructive feedback. We appreciate that all four reviewers provide positive overall evaluations and indicate that our work can be accepted.

Four reviewers all agree that our work is the first to integrate imitation learning and reinforcement learning in a 3DGS-based visual simulation environment with end-to-end policy learning, and recognize our contribution to improving visual fidelity in autonomous driving simulation.

Two reviewers (BysM and 3GxG) have confirmed that their concerns are fully addressed. Reviewer uDLu noted that, although their main concerns have been resolved, they remain unsure whether low-level metrics such as collision rate can reliably assess performance. We have clarified that we employ a low-level discretized action space to reduce the gap between predicted trajectories and actual control commands. This approach enables closed-loop performance to more accurately reflect the true capabilities of the model, thereby making low-level metrics such as collision rate both meaningful and reliable. To further ensure the reliability of collision metrics, we perform precise geometric checks within the 3D environment and apply multi-frame temporal filtering to mitigate transient detection errors.

Regarding the concerns raised by reviewers kB6h and BysM about reproducibility and openness, we will release our code as planned to ensure reproducibility and support future research. The code is being prepared for public release.

Thank you again for the time and consideration.

---

### Decision · Program_Chairs · 2025-09-17

**Decision:**

Accept (poster)

**Comment:**

The paper proposes a reinforcement learning framework for end-to-end autonomous driving. It integrates imitation learning as regularization, uses a photorealistic digital twin for closed-loop training, and designs safety-aware rewards. Results show significant improvements, particularly lower collision rates, over IL baselines.

Strengths: first integration of IL+RL in 3DGS photorealistic environments; extensive closed-loop evaluation across thousands of scenarios; strong safety performance (3x lower collision rate); clear pipeline and ablations; code planned for release.

Weaknesses: limited novelty beyond combining existing components; reliance on a self-created benchmark limits external comparability; high computational cost; missing discussion on sim-to-real transfer; reproducibility dependent on future code release.

During rebuttal, authors added evaluations under diverse weather/road conditions, clarified IL+RL distinctions vs prior work, explained decoupled action space, addressed reward hacking, and confirmed open-sourcing plans. Reviewers acknowledged clarifications, with two raising scores and all maintaining positive recommendations. Concerns on novelty and benchmark remain but were judged minor compared to contributions.